# Upscaling instantaneous to daily evapotranspiration using modelled daily shortwave radiation for remote sensing applications: an Artificial Neural Network approach

Loise Wandera[1,2], Kaniska Mallick[1], Gerard Kiely[3], Olivier Roupsard[4], Matthias Peichl[5], Vincenzo Magliulo[6]

[1]Remote Sensing and Ecohydrological Modeling, Dept. ERIN, Luxembourg Institute of Science and Technology, Belvaux, Luxembourg

[2]Water Resources, Dept. ITC, University of Twente, Enschede, Netherlands

[3]Hydrology, Micrometeorology & Climate Investigations, HYDROMET Research Group, University College Cork, Ireland

[4]CIRAD, UMR Ecology and Soils, Montpellier, France

[5]Forest Landscape Biogeochemistry, Dept. Swedish University of Agricultural Sciences Umeå, Sweden

[6]Consiglio Nazionale delle Ricerche, ISAFOM, Ercolano, Napoli, Italy

Correspondence to: Kaniska Mallick (Phone: +352 275888425; email: kaniska.mallick@gmail.com); Loise Wandera (email: loise.wandera@list.lu);

## 1 Abstract

Upscaling instantaneous evapotranspiration retrieved at any specific time-of-day ($ET_i$) to daily
evapotranspiration ($ET_d$) is a key challenge in mapping regional $ET$ using polar orbiting
sensors. Various studies have unanimously cited the short wave incoming radiation ($R_S$) to be
the most robust reference variable explaining the ratio between $ET_d$ and $ET_i$. This study aims
to contribute in $ET_i$ upscaling for global studies using the ratio between daily and
instantaneous incoming short wave radiation ($R_{Sd}/R_{Si}$) as a factor for converting $ET_i$ to $ET_d$.
This paper proposes an artificial neural network (ANN) machine learning algorithm first to
predict $R_{Sd}$ from $R_{Si}$ followed by using the $R_{Sd}/R_{Si}$ ratio to convert $ET_i$ to $ET_d$ across different
terrestrial ecosystem. Using $R_{Si}$ and $R_{Sd}$ observations from multiple sub-networks of
FLUXNET database spread across different climates and biomes (to represent inputs that
would typically be obtainable from remote sensors during the overpass time) in conjunction
with some astronomical variables (e.g., solar zenith angle, day length, exoatmospheric
shortwave radiation etc.), we developed ANN model for reproducing $R_{Sd}$ and further used it to
upscale $ET_i$ to $ET_d$. The efficiency of the ANN is evaluated for different morning and
afternoon time-of-day, under varying sky conditions, and also at different geographic
locations. $R_S$-based upscaled $ET_d$ produced a significant linear relation ($R^2$ = 0.65 to 0.69),
low bias (-0.31 to -0.56 MJ m$^{-2}$ d$^{-1}$) (appx. 4%), and good agreement (RMSE 1.55 to 1.86 MJ
m$^{-2}$ d$^{-1}$) (appx. 10%) with the observed $ET_d$, although a systematic overestimation of $ET_d$ was
also noted under persistent cloudy sky conditions. Inclusion of soil moisture and rainfall
information in ANN training reduced the systematic overestimation tendency in
predominantly overcast days. An intercomparison with existing upscaling method at daily, 8-
day, monthly, and yearly temporal resolution revealed a robust performance of the ANN
driven $R_S$-based $ET_i$ upscaling method and was found to produce lowest RMSE under cloudy
conditions. Sensitivity analysis revealed variable sensitivity of the method to biome selection
and high $ET_d$ prediction errors in forest ecosystems are primarily associated with greater
rainfall and clouds. The overall methodology appears to be promising and has substantial
potential for upscaling $ET_i$ to $ET_d$ for field and regional scale evapotranspiration mapping
studies using polar orbiting satellites.
*Key Words*: Evapotranspiration, upscaling, artificial neural networks, short wave radiation,
rainfall, soil moisture, FLUXNET
**1 Introduction**
Satellite-based mapping and monitoring of daily regional evapotranspiration (*ET* hereafter)
(or latent heat flux, $\lambda E$) is considered as a key scientific concern for multitudes of applications
including drought monitoring, water rights management, ecosystem water use efficiency
assessment, distributed hydrological modelling, climate change studies, and numerical
weather prediction (Anderson et al., 2015; Senay et al., 2015; Sepulcre-Canto et al., 2014). *ET*
variability during the course of a day is influenced by changes in the radiative energy being
received at the surface (Brutsaert & Sugita, 1992; Crago, 1996; Parlange & Katul, 1992), due
to soil moisture variability particularly in the water deficit landscapes, and also due to the
stomatal regulation by vegetation.
One of the fundamental challenges in regional *ET* modelling using polar orbiting satellites
involves the upscaling of instantaneous *ET* retrieved at any specific time-of-day ($ET_i$
hereafter) to daily *ET* ($ET_d$ hereafter). For example, $ET_i$ retrieved from LANDSAT, ASTER
and MODIS sensors typically represent $ET_i$ at a single snapshot of 1000, 1030 and 1330 hrs
local time, which needs to be upscaled to daily timescale for making this information usable
to hydrologists and water managers (Cammalleri et al., 2014; Colaizzi et al.,  2006; Ryu et al.,
2012; Tang et al., 2013).
In order to accommodate the temporal scaling challenges encountered by remote sensing
based *ET* models, techniques have been proposed and applied by various researchers to
upscale $ET_i$ to $ET_d$. These include:  (1) the constant evaporative fraction (*EF*) approach which
assumes a constant ratio between $\lambda E$ and net available energy ($\phi = R_n - G$, $R_n$ is the net
radiation and *G* is the ground heat flux) during daytime [$EF = \lambda E/(R_n - G)$] (Gentine et al.,
2007; Shuttleworth et al., 1989), (2) constant reference evaporative fractions ($EF_r$) method
where the ratio of $ET_i$ between a reference crop (typically grass measuring a height of 0.12 m
in an environment that is not water limited) and an actual surface is assumed to be constant
during daytime, allowing $ET_d$ to be estimated from the daily $EF_r$ (Allen et al.,1998; Tang et
al., 2013), (3) constant global shortwave radiation method ($R_S$) where $R_S$ is the reference
variable at the land surface and it is assumed that the ratio of daily to instantaneous shortwave

radiation ($R_{Sd}$ and $R_{Si}$) values (i.e., $R_{Sd}/R_{Si}$) determines $ET_d$ to $ET_i$ ratio (Jackson et al., 1983; Cammalleri et al., 2014), and (4) constant extra-terrestrial radiation method where the exo-atmospheric shortwave radiation ($R_STOA$) is the reference variable and the ratio of instantaneous to daily $R_STOA$ ($R_{Si}TOA$ and $R_{Sd}TOA$) is assumed to determine the ratio of $ET_d$ to $ET_i$ (Ryu et al., 2012; Van Niel et al., 2012). These methods have been reviewed and compared in different studies with the view of identifying the most robust $ET_i$ to $ET_d$ upscaling approach based on different data sets, time integrals and varying sky conditions (Cammalleri et al., 2014; Ryu et al., 2012; Tang et al, 2013, 2015; Van Niel et al., 2012; Xu et al., 2015).

Based on the previous studies, we find that the $R_STOA$ approach performed consistently good at lower temporal resolution namely eight-day to monthly scales (Ryu et al., 2012; Van Niel et al., 2012) as well as under clear-sky conditions (Cammalleri et al., 2014), whereas the $R_S$ approach was identified as the most preferred method for $ET_i$ to $ET_d$ conversion at a higher temporal scale i.e. daily timescale in addition to under variable sky conditions (Cammalleri et al., 2014; Chávez et el., 2008; Colaizzi, et al., 2006; Xu et al., 2015). Although the $EF_r$-based method produced comparable $ET_d$ estimates as the $R_S$-based method, however the dependence of $EF_r$ estimates on certain variables (e.g., daily net available energy; $\phi$ and wind speed) and the difficulty to characterise them at the daily scale from single acquisition of polar orbiting satellites (Tang et al., 2015) makes it a relatively less attractive method. Furthermore the $EF$-based method appeared to consistently underestimate $ET_d$ in all these studies.

The motivation of the current work is built on the conclusions of Colaizzi et al. (2006), Chávez et al. (2008), Cammalleri et al. (2014), and Xu et al. (2015) that the ratio of the instantaneous to daily $R_S$ incident on land surface is the most robust reference variable explaining the ratio between $ET_d$ and $ET_i$ among all the tested methods. This work aims to contribute in $ET_i$ upscaling by first developing a method for estimating $R_{Sd}$ from any specific time-of-day $R_S$ information ($R_{Si}$) and further using $R_{Sd}/R_{Si}$ ratio as a factor for converting $ET_i$ to $ET_d$. We develop an artificial neural network (ANN) machine learning algorithm (McCulloch & Pitts, 1943) for estimating $R_{Sd}$. Although net radiation ($R_N$) is more closely associated with $ET$, but $R_S$ constitutes 80-85% of $R_N$ (Mallick et al., 2015). Also from remote sensing perspective, $R_{Si}$ is relatively easily retrievable irrespective of the sky conditions (Wang et al., 2015; Lopez and Batlles, 2014), and its relationship to $R_{Sd}$ is primarily governed

by cloudiness (cloud fraction, cloud optical depth) and astronomical variables (e.g., solar zenith angle, day length, $R_S TOA$ etc.). Given the information of cloudiness is also obtainable from remote sensing, we consider $R_S$ to be a robust variable to explore $ET_i$ upscaling.

Even though this study is intended for remote sensing application, we tested the method using meteorological and surface energy balance flux measurements from eddy covariance (EC) system at the FLUXNET (Baldocchi et al., 2001) sites mainly for the purpose of temporal consistency. However, we evaluate the performance in consideration with overpass time of polar orbiting satellites commonly used in operational $ET$ mapping namely MODIS and LANDSAT. By choosing to use data distributed over different ecosystems and climates zones, we are faced with two problems : (1) changing cloud conditions across ecosystems, (2) varying energy balance closure (EBC) requirements for the fluxes in different ecosystems (Foken et al., 2006; Franssen et al., 2010; Mauder & Foken, 2006; Wilson et al., 2002). Currently, information on cloudiness is obtainable from geostationary meteorological satellites, at hourly to 3-hourly time steps e.g., from the Clouds and Earth's Radiant Energy System (CERES), the International Satellite Cloud Climatology Project–Flux Data (ISCCP-FD), and Global Energy and Water cycle Experiment Surface Radiation Budget (GEWEX-SRB). The CERES algorithm uses cloud information from MODIS onboard both Terra and Aqua platforms and combines it with information from geostationary satellites to accurately capture the diurnal cycles of clouds. In this study, cloudiness is not included in the list of variables used to estimate $R_{Sd}$ due to inconsistency in spatial resolution of data to match with the other predictive variables used. Including cloudiness holds a great potential in improving the ANN $R_{Sd}$ predications due to their direct relationship (Mallick et al., 2015). However, we assess the performance of the ANN under cloudy sky conditions based on simple cloudiness index computations as adopted from previous works (Baigorria et al., 2004). The EBC problems have been reported to vary across landscapes due to management practices, climate, seasons and plant functional type characteristics (Foken et al., 2006). In this study, in order to test the robustness of the proposed method, we initially disregard the site specific EBC problems and assume that the systematic bias of fluxes fall within the same range across entire FLUXNET database used.

The objectives of the present study are: (1) using a ANN with Multilayer Perceptron (MLP) architecture to predict $R_{Sd}$ based on $R_{Si}$ satellite observations, (2) applying $R_{Sd}/R_{Si}$ ratio as a

scaling factor to upscale $ET_i$ to $ET_d$ under all sky conditions, and (3) comparing the
performance of proposed $R_S$-based $ET_i$ upscaling method with $R_S TOA$ and $EF$-based $ET_i$
upscaling methods across a range of temporal scales, biomes and variable sky conditions.
**2  Methodology**
**2.1 Rationale**
The presented method of $ET$ upscaling from any specific time-of-day to daytime average
evaporative fluxes is based on the assumption of self-preservation of incoming solar energy
(i.e., shortwave radiation) as proposed by Jackson et al. (1983).
$$ET_d \approx ET_i \frac{R_{Sd}}{R_{Si}} \qquad (1)$$
Where, $ET_d$ is the daily average evapotranspiration in W m$^{-2}$, $ET_i$ is the instantaneous
evapotranspiration at any instance during daytime in W m$^{-2}$, $R_{Si}$ and $R_{Sd}$ are the values of
shortwave radiation recorded at any instance and the daily average having units W m$^{-2}$. Daily
total $ET_d$ and $R_{Sd}$ is expressed in MJ m$^{-2}$ d$^{-1}$ by using standard conversion from Watts to Mega
Joules. Following Jackson et al. (1983) and Cammalleri et al. (2014), we hypothesized that
the mean diurnal variation of $ET$ for any particular day scales with the mean diurnal variation
of $R_S$. The justifications are: (a) $R_S$ is the principal driver that controls sub-daily $ET$ variability
unless there is substantial diurnal asymmetry in cloudiness or abrupt change in sub-daily soil
moisture between morning and afternoon. (b) Under thick cloudy conditions, $ET$ scales with
$R_S$. Under clear sky conditions $ET$ also scales with $R_S$ and both are in phase if sufficient soil
moisture is available at the surface. (c) Phase difference between $R_S$ and $ET$ are commonly
found under soil moisture deficit conditions in clear-sky days. However, the magnitude of
clear-sky $ET_i$ in water deficit conditions is also be very low, which will lead to substantially
low $ET_i/R_{Si}$ ratio, and would unlikely to introduce any uncertainty in $ET_i$ to $ET_d$ upscaling in
the framework of eq. (1).
For any remote sensing studies using polar orbiting satellites, although the retrieval of $ET_i$ and
$R_{Si}$ has been standardised (Tang et al., 2015; Huang et al., 2012; Polo et al., 2008; Laine et al.,
1999), but, estimating $R_{Sd}$ and $ET_d$ from $R_{Si}$ and $ET_i$ are still challenging. Presently, upscaling
$R_{Si}$ to $R_{Sd}$ is primarily based on the clear sky assumption, i.e., for the entire daytime

integration period, the sky remains cloud-free (Bisht et al., 2005; Jackson et al., 1983). However, the clear-sky assumption is not always appropriate for upscaling remote sensing based $R_{Si}$ and hence $ET_i$ because the sky conditions during a specific time-of-day may be clear whereas the other part of the day might be cloudy. Under such conditions, the clear-sky assumption of $ET_i$ upscaling will lead to substantial overestimation of $ET_d$ in cloudy conditions. Hence reliable estimates of all-sky (i.e., both clear and cloudy) $R_{Sd}$ would greatly improve the $ET_d$ estimates in the framework of eq. (1). Given the unavailability of a definite method to directly estimate all-sky $R_{Sd}$ from $R_{Si}$ information, here we proposed a simple method to upscale $R_{Si}$ to $R_{Sd}$ using ANN. This method uses the observations of both $R_{Sd}$ and $R_{Si}$ from all the available FLUXNET sites in conjunction with some ancillary variables to build the ANN as described in section 2.2. A schematic diagram of the ANN method is given in Fig. 1. The analysis is based on 24-hour period, meaning night time $ET$ contribution is implicitly considered. However, studies have already shown that the nighttime $ET$ in semi-arid and sub-humid regions contributes only $2 - 5\%$ of the total season $ET$ (Malek, 1992; Tolk et al., 2006), and therefore does not appear to be significant.

The overarching aim of this study is to develop an approach that would help in the upscaling of $ET_i$ (retrieved at satellite overpass time) to $ET_d$. Additional value of this study also consists of exploiting $R_{Si}$ information at satellite local crossing time to predict $R_{Sd}$ which is not directly retrievable from any polar orbiting satellites, so that the ratio of $R_{Sd}/R_{Si}$ can be further used to upscale $ET_i$ to obtain $ET_d$ estimates. Currently we are limited to demonstrating with MODIS satellite overpass times (Terra and Aqua), however for the future missions with different local overpass time, the method would still be applicable.

In any natural ecosystem, $R_S$ on a particular day is primarily influenced by the cloud (especially cloud cover fraction and optical thickness) (Mallick et al., 2015; Hildebrandt et al., 2007), latitude, season, and time-of-day. Therefore, $R_{Sd}$ on any specific day is expected to be a function of $R_{Si}$ (as a representative of $R_S$ and cloudiness factors), solar zenith angle (representing latitude, season, time-of-day), day length (representing latitude and season), and $R_S TOA$ (representing latitude, season, time-of-day). Besides, atmospheric aerosols also interact with $R_S$ and absorb some of the radiation particularly in the urban areas. Considering the applications of $ET_i$ to $ET_d$ modeling in the natural ecosystems, we include $R_{Si}$, $R_{Si}TOA$, $R_{Sd}TOA$, solar zenith angle and day length for $R_{Sd}$ (and subsequently $ET_d$) prediction.

**2.2 Development of Artificial Neural Network (ANN)**

ANN is a non-linear model which works by initially understanding the behaviour of a system based on a combination of a given number of inputs and subsequently is able to simulate the system when fed with independent set of inputs of the same system. ANN approach has been successfully used in estimating global solar radiation in many sectors and more so in the field of renewable energy (Ahmad et al., 2015; Hasni et al., 2012; Lazzús et al., 2011). Multi-layer perceptron (MLP) is one of the ANN architectures commonly used as opposed to other statistical methods, makes no prior assumptions concerning the data distribution, has ability to reasonably handle non-linear functions and reliably generalise independent data when presented (Gardner & Dorling, 1998; Khatib, Mohamed, & Sopian, 2012; Wang, 2003). In the present study, MLP was chosen as it has been widely used in many similar studies and cited to be a better alternative as compared to the conventional statistical methods (Ahmad et al., 2015; Chen et al., 2013; Dahmani et al., 2016; Mubiru & Banda, 2008). The MLP is composed of 5 neurons in the input layer, 1 output layer and 10 hidden layers (Fig. 2). The input layer neurons are made up of instantaneous incoming short wave radiation ($R_{Si}$), instantaneous exo-atmospheric shortwave radiation ($R_{Si}TOA$), daily exo-atmospheric shortwave radiation ($R_{Sd}TOA$), solar zenith angle ($\theta_Z$), and day length ($L_D$) as the predictor variables whose values are initially standardized to range between -1 to 1. The choice of the inputs is intentionally limited to the variables that cannot only be acquired by measurements from meteorological stations but also derived from simple astronomical computations (Ryu et al., 2012) mainly to help minimize on the spatial distribution problem (as described earlier in the introduction) that is often linked to ground weather stations. In the MLP processing, the input layer directs the values of each input neuron $x_i$ ($i$ = 1, 2, 3…. $n$) into each neuron ($j$) of the hidden layers. In the hidden layer, $x_i$ is multiplied by a weight ($w_{ij}$) followed by a *bias* ($b_j$) assigned for each hidden layer also is applied. The weighted sum (eq. (2)) is fed into a transfer function. In this work a tangent sigmoid (TANSIG) function is used (eq. (3)) in the hidden layer while in the output layer a PURELIN function is applied (eq. (4)) to give a single output value which is the predicted daily shortwave radiation ($R_{Sd\_pred}$). PURELIN is a linear neural transfer function used in backpropagation network. It calculates a layer's output from its net input. The function generates outputs between zero and 1 as the neuron's net input goes from negative to positive infinity. The training of the ANN is completed by a regression

analysis being performed internally by the algorithm between the target variable i.e. the
observed and predicted daily shortwave radiation ($R_{Sd\_obs}$ and $R_{Sd\_pred}$).

$$x_j = \int \left( \sum_{i=1}^{n} w_{ij}\, y_i\, b_j \right) \tag{2}$$

$$y_j = \frac{2}{(1 + \exp(-2X_i) - 1)} \tag{3}$$

$$y_j = X_i \, (PURELIN) \tag{4}$$

Bayesian regularization algorithm was chosen for the optimization process because it is able
to handle noisy datasets by continuously applying adaptive weight minimization and can
reduce or eliminate the need for lengthy cross-validation that often leads to overtraining and
overfitting of models (Burden and Winkler, 2009).
**2.3 Datasets**
Daily and half-hourly data on $R_S$ (W m$^{-2}$), $R_{STOA}$, net radiation ($R_n$, W m$^{-2}$), latent heat flux
($\lambda E$, W m$^{-2}$), sensible heat flux ($H$, W m$^{-2}$) and ground heat flux ($G$, W m$^{-2}$) measured by the
FLUXNET (Baldocchi et al., 2001) eddy covariance network were used. A total of 126 sites
from the years 1999 to 2006 distributed between latitude 0-90 degrees north and south of the
equator were used for the present analysis. The data sites covered a broad spectrum of
vegetation functional types and climatic conditions and a list of the sites are given in Table S1
in the supplementary section.
Among 126 sites, 85 sites were used for training and remaining 41 sites were used for
validation. Partition of the data into training and validation was randomly selected regardless
of the year. These translated into 194 and 86 yearly data for the respective sample. A global
distribution of the data sites is shown in Fig. 3. From the training dataset, three samples were
internally generated by the algorithm i.e., training datasets, validation datasets, and a testing
dataset in a percentage ratio of 80:15:5 respectively. The ANN algorithm is designed to
validate its performance for any given training which in most cases should be sufficient for
validating the network. However to ensure the network is robust, we further test the generated
network with independent dataset. Considering the equatorial crossing time of different polar
orbiting sensors like LANDSAT, ASTER, and MODIS Terra-Aqua, unique networks were
generated for different time of day from morning to afternoon, and thus we had a total of 8
networks to represent potential satellite overpass times between 1030 to 1400 hours using 30
minutes interval as the closest reference time for each hour. The generated networks were
then applied to an independent validation data set.
**2.4 Intercomparison of $ET_i$ upscaling methods**
An intercomparison of three different $ET_i$ upscaling methods is performed with the
homogeneous datasets to assess their relative performance across a range of temporal scales
and variable sky conditions. These are: (a) $R_S$-based upscaling method, where ANN predicted
$R_{Sd}$ is used in conjunction with observed $R_{Si}$ to predict $ET_d$ using eq. (1).
(b) The exo-atmospheric irradiance method (Ryu et al., 2012) where the reference variable is
$R_S TOA$.

$$R_{Sd}TOA = S_{sc}\left[1 + 0.033cos\left(\frac{2\pi t_d}{365}\right)\right]cos\theta_Z \qquad (5)$$

$$SF_{RTOA} = \frac{R_{Sd}TOA}{R_{Si}TOA} \qquad (6)$$

$$ET_d = ET_i SF_{RTOA} \qquad (7)$$

Where $S_{sc}$ is the solar constant (1360 W m$^{-2}$), $t_d$ is the day of year (DoY), and $\theta_Z$ is the solar
zenith angle.
(c) $EF$-based method (Cammalleri et al., 2014), where reference variable is the net available
energy ($\phi$) (i.e., $R_n$ - $G$).

$$SF_{EF} = \frac{ET_i}{(R_n - G)_i} \qquad (8)$$

$$ET_d = 1.1(R_n - G)_d SF_{EF} \qquad (9)$$

Where $SF_{EF}$ is the $EF$-based scaling factor, $(R_n - G)_i$ and $(R_n - G)_d$ are the instantaneous and
daily net available energy, respectively.
We tested the performance of the three upscaling algorithms for all possible sky conditions
assumed to be represented by daily atmospheric transmissivity ($\tau_d$) (eq. 10) namely (i)
$0.25 \geq \tau \geq 0$ ($\tau_1$, hereafter), (ii) $0.5 \geq \tau \geq 0.25$ ($\tau_2$, hereafter) (iii) $0.75 \geq \tau \geq 0.5$ ($\tau_3$, hereafter), and (iv)
$1 \geq \tau \geq 0.75$ ($\tau_4$, hereafter), respectively. We use daily $\tau$ because it indicates the overall sky
condition throughout a day.

$$\tau_d = \frac{R_{Sd}}{R_{Sd}TOA} \tag{10}$$

$R_{Sd}$ and $R_{Sd}TOA$ are daily shortwave radiation and the exo-atmospheric shortwave radiation in
MJ m$^{-2}$ d$^{-1}$ (converted from W m$^{-2}$).
**2.5 Statistical error analysis**
The relative performance of the ANN and three upscaling methods is evaluated using
statistical indices generated namely: coefficient of determination ($R^2$), root mean square error
(RMSE), mean absolute percentage error (MAPE), index of agreement (IA), and bias. $ET_d$
estimates using the respective upscaling coefficients were compared with measured $ET_d$.

$$R^2 = 1 - \frac{\sum_{i=1}^{n}(p_i - o_i)^2}{\sum_{i=1}^{n}(o_i)^2} \tag{11}$$

$$RMSE = \sqrt{\frac{\sum_{i=1}^{n}(o_i - p_i)^2}{n}} \tag{12}$$

$$MAPE = \frac{1}{n}\sum_{i=1}^{n}\frac{|o_i - p_i|}{n} * 100 \tag{13}$$

$$IA = \frac{\sum_{i}^{n}(p_i - o_i)^2}{\sum_{i=1}^{n}(|p_i - o_i| + |o_i - p_i|)^2} \tag{14}$$

$$Bias = \frac{\sum_{i=1}^{n}(p_i - o_i)}{n} \tag{15}$$

Where, $n$ is the number of data points; $o_i$ and $p_i$ are daily observed and estimated $R_{Sd}$ or $ET_d$,
respectively. $\bar{O}$ was the mean value of observed $R_{Sd}$ or $ET_d$.
**2.6 Sensitivity of ANN training and validation**
Given the majority of the FLUXNET sites represent forest biomes and the distribution of EC
sites over non-forest biomes are proportionately lower as compared to the forests, we
performed a sensitivity analysis of the ANN-based approach by assessing the error statistics
($R^2$ and RMSE) of predicted $ET_d$ for different scenarios of ANN training. Three case studies
were generated: (a) Case1, where ANN was trained by including data randomly from the
forests and $ET_d$ validation was done in non-forest biomes (i.e., grassland, crops and
shrublands); (b) Case2, where ANN was trained by including data randomly from the non-
forest biomes and predicted $ET_d$ was evaluated in forest biome; (c) ANN was trained by using
data randomly from equal proportions of forest and non-forest biomes, and $ET_d$ validation was
also done in forest and non-forest biomes. Each individual case was replicated 10 times and
an ensemble mean statistics of predicted $ET_d$ is reported in section 3.5.
**3  Results and discussion**
**3.1 Testing the performance of predicted $R_{Sd}$**
Given that the performance of $ET_d$ upscaling depends on the soundness of $R_{Sd}$ estimation, we
first evaluate the efficacy of the ANN method for predicting $R_{sd}$. Figure 4 summarises the
statistical results of predicted $R_{Sd}$ ($R_{Sd\_pred}$, hereafter) including all the site-year average $R^2$,
RMSE, IA, and MAPE values for eight different time-of-day upscaling time slots. The RMSE
of $R_{Sd\_pred}$ from forenoon upscaling varied between 1.81-1.85 MJ m$^{-2}$ d$^{-1}$, with MAPE, $R^2$, IA
varying between 20–21%, 0.76–0.77, and 0.79 and 0.80, respectively (Fig. 4). For the
afternoon, these statistics were almost similar and varied between 1.83–1.96 MJ m$^{-2}$ d$^{-1}$, 19-
20%, 0.75–0.77, and 0.80–0.81 (Fig. 4). Given the minimal discrepancy in error statistics
from both forenoon and afternoon integration and considering the MODIS Terra-Aqua
average overpass time we have considered 1100 and 1330 hours of daytime for the detailed
follow up analysis.
Figure 5 (a, b) evaluates $R_{Sd\_pred}$ statistics under different level of atmospheric transmissivity
($\tau$) ($0.25 \geq \tau \geq 0$, $0.5 \geq \tau \geq 0.25$, $0.75 \geq \tau \geq 0.5$, and $1 \geq \tau \geq 0.75$) with an overall RMSE of 1.81 and
1.83 MJ m$^{-2}$ d$^{-1}$ for the forenoon and afternoon upscaling respectively. Table 1 and Fig. 5
clearly show an overestimation tendency of the current method under persistent cloudy sky
conditions ($\tau_1$), whereas the predictive capacity of the ANN model is reasonably strong with
increasing atmospheric clearness. The RMSE of $R_{Sd\_pred}$ for different $\tau$ class from forenoon
upscaling varied between 0.62 to 2.45 MJ m$^{-2}$ d$^{-1}$, with MAPE, R$^2$ and IA of 9.2 to 53%, 0.67
to 0.98, and 0.67 to 0.95, respectively (Table 1). For the afternoon upscaling these statistics
were 0.89 to 2.4 MJ m$^{-2}$ d$^{-1}$ (RMSE), 2.4 to 52% (MAPE), 0.65 to 0.98 (R$^2$), and 0.67 to 0.95
(IA) (Table 1).
The overestimation of $R_{Sd\_pred}$ at low values of $\tau$ is presumably associated with varying levels
of cloudiness during the daytime. Since $R_{Sd\_pred}$ depends on the magnitude of $R_{Si}$, $L_D$, $\theta_Z$,
$R_{SiTOA}$, and $R_{SdTOA}$, there will be a tendency of overestimating $R_{Sd\_pred}$ on partly cloudy days if
$R_{Si}$ at a specific time-of-day is not affected by the clouds ($L_D$, $\theta_Z$, $R_{SiTOA}$, and $R_{SdTOA}$ are not
influenced by the clouds).
**3.2 Evaluation of predicted $ET_d$ based on $R_{Sd\_pred}$**
Figure 6 summarises the statistical results of predicted $ET_d$ ($ET_{d\_pred}$, hereafter) for eight
different time-of-day slots. Upon statistical evaluation, all the cases showed significantly
linear relationship between $ET_{d\_pred}$ and observed $ET_d$ ($ET_{d\_obs}$, hereafter). The RMSE of
$ET_{d\_pred}$ from forenoon upscaling varied from 1.67–1.84 MJ m$^{-2}$ d$^{-1}$, with MAPE, R$^2$, IA
varying between 30%–34%, 0.62–0.68, and 0.77–0.80, respectively (Fig. 6). For the afternoon
upscaling, these statistics varied between 1.5–1.6 MJ m$^{-2}$ d$^{-1}$, 29%–30%, 0.67–0.71, and 0.80
(Fig. 6). These results also indicate that the error statistics were nearly uniform and the
accuracy of $ET_{d\_pred}$ varied only slightly when integration was done from different time-of-
day hours between 1030 to 1400 h. These typical error characteristics can greatly benefit the
$ET_d$ modelling using polar orbiting data with varying overpass times between 1030 to 1400
hours. This also opens up the possibility to use either forenoon satellite (e.g., MODIS Terra,
LANDSAT, ASTER etc.) or afternoon satellite (i.e., MODIS Aqua) to upscale $ET_i$ to $ET_d$.
Following $R_{Sd}$, here also we restricted our analysis to the two different time-of-day (1100h
and 1330h) representing Terra and Aqua overpass times.
Figure 7 (a and b) compares $ET_{d\_pred}$ against $ET_{d\_obs}$ for different level of daily $\tau$. The overall
RMSE, MAPE, and $R^2$ were 1.86 and 1.55 MJ m$^{-2}$ d$^{-1}$, 31% and 36%, 0.65 and 0.69 for the
forenoon and afternoon upscaling, respectively. As seen in Fig. 7, there is a systematic
overestimation of $ET_{d\_pred}$ relative to the tower observed values for low range of $\tau$ (i.e., cloudy
sky). It is important to realise that, unlike $ET_{d\_obs}$, $ET_{d\_pred}$ might be an outcome of $ET_i$
instances when the sky was not overcast, i.e., the sky conditions might be clear at specific
time-of-day but can be substantially overcast for the remainder of the daytime. As a result,
any bias in the daily shortwave radiation prediction ($R_{Sd\_pred}$) will result in biased $ET_{d\_pred}$
according to eq. 1, and the omission of non-clear sky conditions at any particular time of
daytime would tend to lead to $ET_{d\_pred} > ET_{d\_obs}$ for generally overcast days. However, there
could be another opposite case that sky is cloudy at e.g., 1100 hr but clear at other times. This
will probably lead to an underestimation of $R_{Sd\_pred}$, and consequently underestimation of
$ET_{d\_pred}$. Such cases were also found in $\tau_3$ categories in Fig. 7 where clouds of data points
clearly falling significantly below the 1:1 line, thus showing substantial underestimation of
$ET_{d\_pred}$. Since $ET_{d\_obs}$ are the integrations of multiple $ET_i$ measurements, such conditions
could be conveniently captured in the observations which were not possible in the current
framework of $ET_{d\_pred}$. Therefore, when upscaling was done under clear skies at nominal
acquisition time for generally overcast days, higher errors in $ET_{d\_pred}$ can be expected
(Cammalleri et al., 2014) and vice-versa. We examined this cloudy sky overestimation pattern
in greater detail by evaluating the error statistics in $ET_{d\_pred}$ for four different levels of daily $\tau$
categories (Fig. 8).
Statistical evaluation of $ET_{d\_pred}$ for different classes of daily $\tau$ (estimated as the ratio between
daily observed $R_{Sd}$ and $R_{Sd}TOA$) indicates the tendency of higher RMSE and low $R^2$ in
$ET_{d\_pred}$ under the persistent cloudy-sky conditions ($\tau_1$), while the performance of $ET_{d\_pred}$ is
reasonably good with increasing atmospheric clearness ($\tau_2$, $\tau_3$, and $\tau_4$) (Fig. 8). The RMSE of
$ET_{d\_pred}$ for different $\tau$ class from forenoon upscaling varied between 1.09 to 2.96 MJ m$^{-2}$ d$^{-1}$,
with MAPE, $R^2$ and IA of 25 to 75%, 0.38 to 0.79, and 0.71 to 0.82, respectively. For the
afternoon upscaling, these statistics were 0.98 to 2.02 MJ m$^{-2}$ d$^{-1}$ (RMSE), 24 to 87%
(MAPE), 0.40 to 0.68 ($R^2$), and 0.71 to 0.77 (IA).

To probe into detail of the high errors under persistent cloudiness conditions, a new ANN was trained by introducing daily precipitation ($P$) and soil moisture ($SM$) information (along with $R_S$, $R_STOA$, $\theta_Z$, and $L_D$) assuming that the inclusion of these two variables might improve the predictive power of $R_S$-based ANN. In the new ANN, we used data from the sites where coincident measurements of $P$ and $S_M$ were available along with $R_S$ and $ET$, and validated $ET_d$ predictions of the new ANN on independent sites. The analysis revealed 34% reduction in RMSE (from 3.28 to 2.88 MJ m$^{-2}$ d$^{-1}$), 16% reduction in MAPE (from 90 to 76%), and 49% reduction in mean bias (0.76 to 0.39 MJ m$^{-2}$ d$^{-1}$) for persistent cloudy-sky cases (i.e., $\tau_1$ scenarios) from 1100 hr upscaling. However, no significant improvements in $ET_{d\_pred}$ were evident for $\tau_2$, $\tau_3$, and $\tau_4$ and also for any of the $\tau$ classes from the afternoon (1330 hr) upscaling (Fig. 9). $ET_d$ is generally controlled by radiation and soil moisture availability. Under the radiation controlled conditions, $ET_d$ is generally not limited due to soil moisture and $70 - 75\%$ of the net radiation is contributed to $ET_d$. Therefore, $R_S$-based method of $ET_i$ upscaling is expected to perform reasonably well unless the upscaling is performed from a clear sky instance for a predominantly overcast or rainy day. However, from Fig. 9 is it apparent that the inclusion of cloud information (cloud fraction, cloud optical thickness) in $R_S$-based ANN would substantially reduce $ET_{d\_pred}$ errors when upscaling is performed from a clear sky instance for a predominantly overcast day and vice-versa. Improvements of $ET_{d\_pred}$ error statistics by including daily $P$ and $SM$ (as an indicator of cloudiness) is also suggestive to the relevance of such approach as a future improvement of the current framework, which is expected to reduce the systematic error under overcast conditions. However, the cloud information available from alternative sources e.g., from the Clouds and Earth's Radiant Energy System (CERES), the International Satellite Cloud Climatology Project–Flux Data (ISCCP-FD), and Global Energy and Water cycle Experiment Surface Radiation Budget (GEWEX-SRB) are available at coarse spatial resolution (100 km$^2$) and combining these information with EC tower measurements to train ANN could also introduce additional errors due to the spatial scale mismatch, is therefore out of scope of the present study.

Figure 10 shows the time series comparisons between observed $ET_d$ and $ET_{d\_pred}$ for four different stations representing different latitude bands of both the Northern (Sweden) and Southern (Brazil, Australia, and South Africa) hemispheres. These reveal that the temporal dynamics of $ET_d$ is in general consistently captured by the proposed method throughout year.

In Br_SP1, relatively less seasonality was found in both observed and predicted $ET_d$. This is because SP1 is a tropical site having an annual rainfall of 850–1100 mm most of which is evenly distributed between March to end of September. The peaks in $ET_d$ values during the beginning of year and October onwards coincided with the periods of increased $R_S$, and $ET_{d\_pred}$ could reasonably capture the observed trends during both rainy and non-rainy periods. Similarly the low $ET_d$ pattern (0.1 to 2 MJ m$^{-2}$ d$^{-1}$) in the hot arid climate of South Africa (Za-Kru) could also be reasonably captured in $ET_{d\_pred}$ (Fig. 10). $ET_{d\_pred}$ in the other Southern hemisphere (AU-Tum) and Northern hemisphere (SE-Fla) sites have shown distinct seasonality (high summer and low winter $ET_d$) coinciding with the observed $ET_d$ patterns.

**3.3 Comparison with existing *ET* upscaling methods**

$ET_{d\_pred}$ from $R_S$-based method was intercompared with two other upscaling schemes ($R_STOA$ and $EF$) over 41 FLUXNET validation sites for two different time-of-day, 1100h and 1330h, the statistics of which are given in Table 2. This comparison was also carried out according to different $\tau$ classes as defined in section 2.2.3.

From Table 2 it is apparent that the $R_S$-based method has generally produced relatively low RMSE (1.21 to 1.99 MJ m$^{-2}$ d$^{-1}$) and MAPE (23 to 50%) as well as relatively high IA (0.72 to 0.84) as compared to $R_STOA$ and $EF$-based upscaling methods. The $EF$-based upscaling method appears to systematically underestimate $ET_d$ for both forenoon and afternoon as evident from high negative bias compared to the other two methods (Table 2). On comparing $R_S$ and $R_STOA$ methods, $R_S$-based method performed relatively better than the $R_STOA$ scheme for low magnitude of $\tau$ (i.e., under predominantly cloudy-sky). However, the results suggest comparable performance of $R_STOA$-based approach under clear sky conditions which are reflected in lowest RMSE (1.09 and 1.13 MJ m$^{-2}$ d$^{-1}$) in $ET_{d\_pred}$ as compared to the other $\tau$ classes. In general, all the schemes performed relatively better from the afternoon upscaling as compared to the morning upscaling (as evidenced in higher R$^2$ and lower bias) (Table 2) which is in agreement with the findings from Ryu et al. (2012). Due to their comparable error statistics, an intercomparison of $R_S$ and $R_STOA$-based methods of $ET_i$ upscaling was also carried out across different biomes.

Biome specific evaluation of $R_S$-based $ET_{d\_pred}$ (Fig. 11) revealed lowest RMSE and highest R$^2$ both in the grassland (GRA) (0.68 to 1.14 MJ m$^{-2}$ d$^{-1}$; 0.53 to 0.79) and shrubland (SH)

(0.66 to 1.76 MJ m$^{-2}$ d$^{-1}$; 0.60 to 0.82) whereas the RMSE was comparatively high over the
tropical evergreen broadleaf forests (EBF) (1.41 to 2.02 MJ m$^{-2}$ d$^{-1}$) and deciduous broadleaf
forests (DBF) (1.94 to 2.55 MJ m$^{-2}$ d$^{-1}$). Similar evaluation with $R_S TOA$-based method
revealed the lowest RMSE and highest R$^2$ in the grassland (0.64 to 1.14 MJ m$^{-2}$ d$^{-1}$; 0.61 to
0.84), and highest RMSE in EBF, DBF, and evergreen needleleaf forests (ENF) (1.57 to 2.05
MJ m$^{-2}$ d$^{-1}$, 1.2 to 2.25 MJ m$^{-2}$ d$^{-1}$ and 0.93 to 4.02 MJ m$^{-2}$ d$^{-1}$) (Fig. 11c and 11d). Higher
$ET_{d\_pred}$ errors in forests are related to the predominant cloudy-sky issue as described earlier.
Tropical evergreen broadleaf forests (and forests in general) have high $ET$, water tends to re-
cycle locally and generate rainfall. Therefore, cloudy sky conditions are more frequent at
tropical evergreen broadleaf forest and other forests types than at grassland and shrublands. In
the biome specific $ET_{d\_pred}$ error statistics (Fig. 11), relatively large bias in crop $ET_{d\_pred}$ is
introduced due to the inclusion of irrigated agroecosystems in the validation. In irrigated
agroecosystems, day-to-day variation in soil moisture is not substantial and $ET_d$ is
predominantly controlled by the net radiation. Therefore, the inclusion of soil moisture in the
current ANN framework is unlikely to improve $ET_{d\_pred}$ statistics in the irrigated
agroecosystems. Further having many explanatory variables (e.g., land management,
irrigation statistics, anthropogenic factors) to train the ANN, we risk overfitting the model and
hence introducing bias. It is also evident that both $Rs$ and $RsTOA$-based method of $ET_d$
estimation would be better suited for natural ecosystem e.g., in the Amazon basin or in the
forest ecosystems where significant hydrological and climatological projections are
emphasizing the role of $ET_d$ to understand the resilience of natural ecosystems in the spectre
of hydro-climatological extremes (Harper et al., 2014; Kim et al., 2012). The performance of
the method in the semi-arid shrublands appear to be promising (Fig. 11) and therefore the
method seems to be credible under water-stressed environment also.
Given this analysis was based on FLUXNET sites distributed across 0-90 degrees latitude
north and south, the training datasets covers substantial climatic and vegetation variability.
The percentage distribution of the training data according to vegetation type was; 23% crops,
31% deciduous broadleaf forest, 10% evergreen broadleaf forest, 20% evergreen need leaf
forest, 8% grassland, 7% shrubs and 1% aquatic as indicated in table S1. The number of
grassland and shrubs as indicated were relatively less as compared to the crops and forests
sites. However, biome specific error statistics (Fig. 11) indicted the absence of any systematic
errors due to vegetation sampling with the exception of EBF. Availability of more EBF sites
in the training datasets is expected to reduce the cloudy-sky errors substantially, due to the
assimilation of more cloud information into the $R_S$-based ANN training.
The tendency of positive bias in $ET_{d\_pred}$ from both $R_S$ and $R_STOA$ in clear skies from
afternoon upscaling is partly explained by the fact that, during the afternoon the values of
both $R_S$ and $R_STOA$ reached maximum limit and dominates their daily values (Jackson et al.,
1983). The post afternoon rate of reduction in $ET$ does not coincide with the shortwave
radiation due to stomatal controls on $ET$, and the total water flux from morning to afternoon
(0700h to 1300h) is generally greater than the total water flux from post afternoon (1500h
onwards) till sunset. Therefore multiplying 1330h $ET_i$ with high magnitude of $R_{Sd}/R_{Si}$ or
$R_{Sd}TOA/R_{Si}TOA$ might lead to an overestimation of $ET_{d\_pred}$ in the clear sky days.
Since extraterrestrial shortwave radiation is not affected by the clouds, $ET_{d\_pred}$ from $R_STOA$
performed comparably with the $R_S$-based $ET_{d\_pred}$ with increasing atmospheric clearness (i.e.,
for the higher levels of daily $\tau$). However, increased differences in the RMSE of $ET_{d\_pred}$
between $R_S$ and $R_STOA$ upscaling in the predominantly cloudy days indicates that more
deviations can be expected in $ET_{d\_pred}$ from these two different method of upscaling under
principally overcast conditions (Tang et al., 2013). This happens because the ratio of $R_{Sd}TOA$
$/R_{Si}TOA$ is not impacted by the clouds and the magnitude of this ratio becomes markedly
different from $R_{Sd}/R_{Si}$ ratio in the presence of clouds, which leads to the differences in $ET_{d\_pred}$
between them. The $R_S$-based method is relatively efficient to discriminate the impacts on $ET$
by $R_{Sd}/R_{Si}$ due to the clouds. The generally good performance of $R_S$-based method and
comparable error statistics with $R_STOA$-based $ET_d$ estimates are consistent with the findings
of Cammalleri et al. (2014) and Van Niel et al. (2012). As shown in Table 2, relatively lower
RMSE of $R_STOA$-based $ET_{d\_pred}$ for atmospheric transmissivity class above 0.75 reveals that
under pristine clear sky conditions $R_STOA$ can be successfully used to upscale $ET_i$. However,
one of the main reasons for the differences in RMSE between $R_S$ and $R_STOA$ method for daily
transmissivity above 0.75 could be due to the fact that if $ET_i$ upscaling is performed from a
cloudy instance for a predominantly clear sky day, then such RMSE difference between the
two different upscaling methods is expected. These results also revealed the probability of a
hybrid $ET_i$ upscaling method by combining cloud information or $SM$ and $P$ in $R_S$-method (for
transmissivity between zero to 0.5) and $R_STOA$-method (for transmissivity greater than 0.5).
However this hypothesis needs to be tested further.
The systematic $ET_d$ underestimation by $EF$-based upscaling method and nearly similar pattern
of bias from two different time-of-day upscaling (Table 2) further points to the fact that the
concave-up shape of $EF$ during daytime (Hoedjes et al., 2008; Tang et al., 2013) will tend to
underestimate $ET_d$ if $EF$ is assumed to be conservative during the daytime. $EF$ remains
conservative during the daytime under extremely dry conditions when $ET_d$ is solely driven by
deep layer soil moisture. The systematic underestimation of $ET_d$ from $EF$-based upscaling
method corroborates with the results reported by other researchers (Cammalleri et al., 2014;
Delogu et al., 2012; Gentine et al., 2007; Hoedjes et al., 2008) which suggests that the self-
preservation of $EF$ is not generally achieved, and this systematic underestimation of $ET_d$ can
be partially compensated if $EF$-based $ET_i$ upscaling is done from morning 0900h or afternoon
1600h time-of-day.
We further resampled $ET_d$ (both predicted and observed) from daily to 8-day, monthly, and
annual scale, and statistical metrics from the three different upscaling methods at three
different temporal scales are shown in Fig. 12 and Table 3. Averaging $ET_d$ at 8-day, monthly
and annual scale substantially reduced the RMSE to the order of 60 to 70% for all the three
upscaling methods. The $R_S$-based upscaled $ET_d$ from morning and afternoon showed reduction
in RMSE from 1.79 MJ to 0.57 MJ and 1.74 MJ to 0.51 MJ from daily to annual $ET$,
respectively. For the other two upscaling method these statistics varied from 1.85 and 1.89 MJ
to 0.62 and 0.53 MJ ($R_STOA$ method), and 2.16 and 1.33 MJ to 2.20 and 1.31 MJ ($EF$
method) (Fig. 12 and Table 3). The impacts of daily cloud variability might have smoothed
out in 8-day, monthly and annual scale which led to reduced RMSE and higher correlation
between $ET_{d\_pred}$ and $ET_{d\_obs}$. Nearly similar error statistics in $ET_{d\_pred}$ from both the morning
and afternoon upscaling also substantiates the findings of Ryu et al. (2012) and greatly
stimulate the use of either morning satellite (i.e., Terra) or after satellite (i.e., Aqua) to upscale
$ET_i$ to $ET_d$ or 8-day mean $ET_d$.
The principal limitation of the approach is the dependence of $ET_d$ and $R_{Sd}$ on single snapshot
of $ET_i$ and $R_{Si}$. Although hourly $R_S$ data from geostationary satellite are becoming available;
but these are available as sectorial products (i.e. for particular continents) instead of full
global coverage. Ongoing efforts to develop geostationary based data by merging multiple
geostationary satellites tend to overcome this limitation.

### 3.4 Impact of energy balance closure on $ET_{d\_pred}$

FLUXNET EC sites have long been identified to be prone to surface energy budget
imbalance, which might lead to (±20%) to (±40%) under measurement of latent heat fluxes.
In order to assess the impacts of surface energy balance (SEB) closure on current $ET_d$
prediction, we further compared the error statistics of $R_S$-based $ET_{d\_pred}$ (Table 4) for both
'closed' and 'unclosed' surface energy balance datasets. These are the subsets of the data
where all the four SEB components ($\lambda E$, sensible heat flux, ground heat flux, and net
radiation) were available and SEB was closed by the residual SEB closure method (Foken,
2006). Table 4 revealed substantially low RMSE (10 to 60%), $R^2$ (8 to 100%) and MAPE (1
to 75%) in $ET_{d\_pred}$ when $ET_i$ upscaling is done by 'unclosed' SEB. A consistently high
positive mean bias (0.63 to 3.83) in $ET_{d\_pred}$ with 'closed' SEB was also noted (Table 4).
Although, various methods exist to close the surface energy balance, but, the impact of
various SEB closure methods on $ET_{d\_pred}$ statistics is beyond the scope of the current study. It
is also important to mention that in the satellite based $ET_i$ retrieval, net available energy is
partitioned into $ET$ and sensible heat flux with the implicit assumption of SEB closure.
Therefore, application of the current ANN framework is expected not to impact the remote
sensing based $ET_i$ to $ET_d$ upscaling. However, for the validation of remote sensing based $ET_d$
retrievals, surface energy balance fluxes from eddy covariance measurements need to be
closed.

### 3.5 Sensitivity of ANN derived $ET_{d\_pred}$ to biome selection

A sensitivity analysis of ANN derived $R_S$-based $ET_{d\_pred}$ revealed variable sensitivity of the
ANN framework to the biome selection. The coefficient of determination ($R^2$) varied between
0.71 to 0.84 and RMSE between 0.96 to 2.10 MJ m$^{-2}$ d$^{-1}$ across three different scenarios of
ANN training and validation (Fig. 13). However, RMSE was found to be relatively high in
forests in Case2, where ANN was trained by using the data from crops, grasslands and
shrublands only. For the Case1 and Case3, no substantial difference was noted (Fig. 13). This
therefore revealed the fact that the inclusion of forests in ANN training leads to lower errors
in $ET_{d\_pred}$ over non-forest biomes, although the reverse scenario in not likely to be true. Since

forests generally have high *ET*, water recycling tends to be more over the forests which produces substantial rainfall, variable atmospheric water vapor, associated cloudiness, and radiation. Cloudiness is a phenomenon that significantly influences the reliability of a model to predict incoming solar radiation as they are directly related to each other. Therefore, when $R_S$-based ANN is trained with data from forests, the model assimilates information on a diverse range of radiative forcings which broaden their applicability in other biomes. This also emphasizes the fact that the performance of such ANN-based approach is primarily sensitive to their training over a broad spectrum of atmospheric conditions.

## 4 Summary and Conclusions

Given the significance of $ET_d$ in remote sensing based water resource management from polar orbiting satellites, this study developed and evaluated a temporal upscaling method for estimating $ET_d$ from different time-of-day instantaneous *ET* ($ET_i$) measurements with the assumption that the ratio between daytime to instantaneous shortwave radiation ($R_{Sd}/R_{Si}$) is the predominant factor governing $ET_d/ET_i$ ratio. However, since $R_{Sd}$ is not directly measurable from the polar orbiting satellites, we trained an ANN with the FLUXNET observations of $R_{Si}$ and $R_{Sd}$, and validated the model to predict $R_{Sd}$ over independent sites, followed by using $R_{Sd}/R_{Si}$ ratio for converting $ET_i$ to $ET_d$. The overarching goal of this study is to provide an operational and robust $ET_i$ upscaling protocol for estimating $ET_d$ from any polar orbiting satellite. The datasets used for the ANN model development covers a wide range of biome, climate, and variable sky conditions. Therefore, we assume the $R_{Sd}$ prediction from ANN to capture a broad spectrum of radiative forcing, which is also reflected in the independent validation of $R_{Sd}$ and $ET_d$ (Fig. 5, Fig. 7, Table 2). However, the performance of this model for satellite retrieval of $R_{Sd}$ (from $R_{Si}$) is dependent on the accuracy of $R_{Si}$ retrieval (Loew et al., 2016). Also, the distribution of sites over the tropics, Africa, and South East Asia are poor, and more sites over these regions are expected to make the ANN model performance more robust.

Based on measurements from 126 flux tower sites, we found $R_S$-based upscaled $ET_d$ to produce a significant linear relation ($R^2 = 0.65$ to 0.69), little bias (-0.31 to -0.56 MJ m$^{-2}$ d$^{-1}$) (appx. 4%), and good agreement (RMSE 1.55 to 1.86 MJ m$^{-2}$ d$^{-1}$) (appx. 10%) with the observed $ET_d$. While the exoatmospheric shortwave radiation driven $ET_i$ upscaling method

(i.e., $R_STOA$-based) appeared to produce slightly lower RMSE (10% lower) under cloud-free
conditions (Table 2), global shortwave radiation driven method (i.e., $R_S$–based method)
demonstrates more robust performance and was found to be better under cloudy conditions.
Despite $R_S$–based method yielded relatively better overall accuracy in $ET_d$ prediction (i.e.,
$ET_{d\_pred}$) statistics when compared with the $R_STOA$ and evaporative fraction based (*EF*-based)
method, statistical analysis of $ET_{d\_pred}$ accuracy of different temporal upscaling methods (as
discussed in section 3.3) suggests that $R_S$ and $R_STOA$ to produce commensurate results under
coarse temporal resolutions (Table 3). Therefore, at the coarse temporal scale (8-day and
above), any of these two methods ($R_S$ and $R_STOA$) can be used for $ET_i$ to $ET_d$ upscaling.
The proposed upscaling method is based on the idea that instantaneous $ET/R_S$ approximates
daily $ET/R_S$, although it implicitly includes the stomatal controls on *ET* observations mediated
by the vegetation. The cases where $ET_i$ is low due to water stress induced strong stomatal
control; low magnitude of *ET* will also be reflected in upscaling $ET_i$ to $ET_d$ (according to eq.
1). However, to account for any carry over effects of the stomatal control on $ET_d$, inclusion of
longwave radiation would likely to improve the scheme. Stomatal control is significantly
dependent on the thermal longwave radiative components, and, therefore, the relative
proportion of downwelling and upwelling longwave radiation is expected to be a stomatal
constraint. However, the availability of longwave radiation measurement stations in the
FLUXNET datasets is limited to formulate ANN and evaluate this hypothesis. In general, the
stomatal and biophysical constraints are imposed in state-of-the-art thermal remote sensing
based $ET_i$ retrieval schemes, and, therefore the ANN framework can be applied to upscale
remote sensing based $ET_i$ to $ET_d$. Also, relatively good performance of the model in semiarid
shrubland also indicated the applicability of the method in water stressed ecosystems where
stomatal controls are predominant.
Among all the upscaling method tested, $R_S$–based method carries maximum information on
the cloudiness and produced generally lowest RMSE, low bias (Table 3), and, therefore,
overall the preferably robust scaling mechanism (at the daily scale) among all the other
methods tested. The true added value of the ANN is for an operational $ET_d$ product from polar
satellites. Currently, the polar Earth orbiting satellites provide us with $ET_i$ only. However, for
most hydrological and ecosystem modeling applications, $ET_d$ is needed. Therefore, for studies
that will opt to apply $R_S$–based method as a scaling algorithm, $R_{Sd}$ will be easily available for

any measurement of $R_{Si}$ by the satellite using the ANN. However, upscaling large-area satellite-based $ET_i$ by using retrieved $R_{Si}$ would require accurate $R_{Si}$ retrieval techniques, which are currently commonplace (Ahmad et al., 2015; Boulifa et al., 2015; Dahmani et al., 2016; Hasni et al., 2012; Li, Tang, Wu, & Liu, 2013) to support regional scale hydrological applications. Of the two other upscaling methods, $R_S TOA$ could be easily applied over large areas, had lower errors than $EF$, had second best RMSE, and overall lowest bias among the two. We conclude that using modelled $R_S$ to upscale $ET_i$ at daily scale appears to be viable for large-area hydrological remote sensing applications from polar orbiting satellites irrespective of any sky conditions.

**Acknowledgements**

The authors thank HiWET (High resolution modelling and monitoring of water and energy transfers in WETland ecosystems) project funded through the Belgian Science Policy (BELSPO) and FNR under the programme STEREOIII (INTER/STEREOIII/13/03/HiWET) (CONTRACT NR SR/00/301). We thank entire FLUXNET site PIs for sharing the eddy covariance data. This work used eddy covariance data acquired by the FLUXNET community and in particular by the following networks: AmeriFlux (U.S. Department of Energy, Biological and Environmental Research, Terrestrial Carbon Program (DE-FG02-04ER63917 and DE-FG02-04ER63911)), AfriFlux, AsiaFlux, CarboAfrica, CarboEuropeIP, CarboItaly, CarboMont, ChinaFlux, Fluxnet-Canada (supported by CFCAS, NSERC, BIOCAP, Environment Canada, and NRCan), GreenGrass, KoFlux, LBA, NECC, OzFlux, TCOS-Siberia, USCCC. We acknowledge the financial support to the eddy covariance data harmonization provided by CarboEuropeIP, FAO-GTOS-TCO, iLEAPS, Max Planck Institute for Biogeochemistry, National Science Foundation, University of Tuscia, Université Laval, Environment Canada and US Department of Energy and the database development and technical support from Berkeley Water Center, Lawrence Berkeley National Laboratory, Microsoft Research eScience, Oak Ridge National Laboratory, University of California–Berkeley and the University of Virginia. LW acknowledges the PhD supervision from Dr. Wout Verhoef and Dr. Christian van der Tol from University of Twente, The Netherlands.

KM designed the analysis, LW performed the research, KM and LW developed the manuscript, and all the coauthors jointly contributed in editing the manuscript. The authors declare no conflict of interest.

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

| Time-of-day (h) | $\tau$ | $R^2$ | RMSE (MJ m$^{-2}$ d$^{-1}$) | IA | MAPE | Bias (MJ m$^{-2}$ d$^{-1}$) |
|---|---|---|---|---|---|---|
| 1100 | $\tau_1$ | 0.67 | 1.84 | 0.67 | 53.56 | 1.12 |
| | $\tau_2$ | 0.79 | 2.45 | 0.80 | 16.69 | 0.59 |
| | $\tau_3$ | 0.88 | 2.30 | 0.82 | 9.17 | -0.74 |
| | $\tau_4$ | 0.98 | 0.63 | 0.95 | 1.69 | 0.08 |
| 1330 | $\tau_1$ | 0.65 | 1.77 | 0.67 | 51.50 | 1.06 |
| | $\tau_2$ | 0.81 | 2.44 | 0.81 | 16.83 | 0.69 |
| | $\tau_3$ | 0.89 | 2.23 | 0.83 | 8.94 | -0.85 |
| | $\tau_4$ | 0.98 | 0.89 | 0.95 | 2.40 | -0.46 |

**Table 2:** A summary of $ET_d$ error statistics by comparing the performance of $R_S$-based, $R_STOA$-based and $EF$-based $ET_i$
upscaling methods with regard to different sky conditions. Here $\tau_1$ represents low atmospheric transmissivity due to high
cloudiness while $\tau_4$ represents high transmissivity under clear sky conditions.

| Time-of-day (h) | $\tau$ | $R^2$ | | | RMSE (MJ m$^{-2}$ d$^{-1}$) | | | IA | | | MAPE | | | Bias (MJ m$^{-2}$ d$^{-1}$) | | |
|---|---|---|---|---|---|---|---|---|---|---|---|---|---|---|---|---|
| | | $R_S$ | $R_STOA$ | $EF$ | $R_S$ | $R_STOA$ | $EF$ | $R_S$ | $R_STOA$ | $EF$ | $R_S$ | $R_STOA$ | $EF$ | $R_S$ | $R_STOA$ | $EF$ |
| 1100 | $\tau_1$ | 0.49 | 0.32 | 0.32 | 1.34 | 1.65 | 2.07 | 0.72 | 0.67 | 0.71 | 50.14 | 66.70 | 64.19 | -0.13 | -0.04 | 0.05 |
| | $\tau_2$ | 0.72 | 0.70 | 0.69 | 1.73 | 1.81 | 1.93 | 0.81 | 0.78 | 0.69 | 26.47 | 32.41 | 36.42 | -0.21 | -0.19 | -0.95 |
| | $\tau_3$ | 0.72 | 0.73 | 0.79 | 1.99 | 1.94 | 2.38 | 0.81 | 0.79 | 0.59 | 24.69 | 25.66 | 40.37 | -0.24 | -0.37 | -1.78 |
| | $\tau_4$ | 0.77 | 0.81 | 0.68 | 1.32 | 1.13 | 2.00 | 0.84 | 0.81 | 0.49 | 32.17 | 30.02 | 55.43 | 0.05 | -0.19 | -1.34 |
| 1330 | $\tau_1$ | 0.52 | 0.34 | 0.29 | 1.21 | 1.68 | 2.34 | 0.73 | 0.69 | 0.71 | 48.29 | 66.09 | 68.14 | -0.11 | 0.08 | 0.12 |
| | $\tau_2$ | 0.73 | 0.72 | 0.71 | 1.71 | 1.93 | 1.86 | 0.82 | 0.79 | 0.71 | 26.12 | 33.71 | 35.33 | -0.01 | 0.24 | -0.88 |
| | $\tau_3$ | 0.75 | 0.75 | 0.76 | 1.89 | 1.96 | 2.43 | 0.82 | 0.82 | 0.61 | 23.17 | 25.82 | 41.65 | 0.09 | 0.14 | -1.75 |
| | $\tau_4$ | 0.79 | 0.86 | 0.80 | 1.32 | 1.09 | 1.86 | 0.84 | 0.86 | 0.49 | 29.54 | 26.59 | 53.91 | 0.10 | 0.11 | -1.38 |


**Table 3:** Error statistics of $ET_{d\_pred}$ at four different temporal scales from three $ET_i$ upscaling methods.

| Time-of-day (h) | Temporal scale | R² | | | RMSE (MJ m⁻² d⁻¹) | | | IA | | | MAPE | | | Bias (MJ m⁻² d⁻¹) | | |
|---|---|---|---|---|---|---|---|---|---|---|---|---|---|---|---|---|
| | | $R_S$ | $R_STOA$ | EF | $R_S$ | $R_STOA$ | EF | $R_S$ | $R_STOA$ | EF | $R_S$ | $R_STOA$ | EF | $R_S$ | $R_STOA$ | EF |
| 1100 | Daily | 0.71 | 0.72 | 0.71 | 1.79 | 1.85 | 2.16 | 0.82 | 0.80 | 0.67 | 28.80 | 32.98 | 57.00 | 0.19 | 0.22 | 1.21 |
| | 8-days | 0.86 | 0.84 | 0.85 | 1.17 | 1.22 | 1.65 | 0.87 | 0.86 | 0.67 | 18.50 | 20.63 | 46.96 | 0.19 | 0.22 | 1.16 |
| | Monthly | 0.89 | 0.88 | 0.88 | 0.99 | 1.04 | 1.61 | 0.89 | 0.67 | 0.67 | 15.52 | 17.22 | 49.72 | 0.19 | 0.22 | 1.16 |
| | Annually | 0.92 | 0.91 | 0.93 | 0.57 | 0.62 | 1.33 | 0.87 | 0.84 | 0.54 | 11.12 | 12.54 | 45.88 | 0.19 | 0.22 | 1.21 |
| 1330 | Daily | 0.75 | 0.74 | 0.69 | 1.74 | 1.89 | 2.2 | 0.83 | 0.82 | 0.67 | 26.59 | 29.89 | 56.45 | -0.04 | 0.17 | -1.18 |
| | 8-days | 0.87 | 0.86 | 0.84 | 1.11 | 1.21 | 1.7 | 0.88 | 0.88 | 0.68 | 16.80 | 17.97 | 50.36 | -0.04 | 0.17 | -1.18 |
| | Monthly | 0.90 | 0.90 | 0.87 | 0.93 | 1.00 | 1.59 | 0.90 | 0.89 | 0.68 | 13.69 | 14.85 | 48.08 | -0.04 | 0.17 | -1.18 |
| | Annually | 0.93 | 0.93 | 0.92 | 0.51 | 0.53 | 1.31 | 0.88 | 0.88 | 0.54 | 9.00 | 9.70 | 44.13 | -0.04 | 0.17 | -1.18 |


**Table 4:** Evaluation of the $R_S$-based ANN predicted $ET_d$ ($ET_{d\_pred}$) error statistics based on 'closed' (EBC) and unclosed' (EBO) surface energy balance under varying sky conditions represented by four different classes of daily atmospheric transmissivity ($\tau$). Here $\tau_1$ represents low atmospheric transmissivity due to high cloudiness while $\tau_4$ represents high transmissivity under clear sky conditions. The statistical metrics of $ET_{d\_pred}$ for two different upscaling hours (1100 and 1330 h) are presented.

| Time-of-day (h) | $\tau$ | $R^2$ | | RMSE (MJ m-2 d-1) | | IA | | MAPE | | Bias (MJ m-2 d-1) | |
|---|---|---|---|---|---|---|---|---|---|---|---|
| | | EBO | EBC | EBO | EBC | EBO | EBC | EBO | EBC | EBO | EBC |
| 1100 | $\tau_1$ | 0.37 | 0.17 | 2.96 | 3.31 | 0.71 | 0.57 | 87.21 | 86.49 | 0.66 | 1.12 |
| | $\tau_2$ | 0.68 | 0.54 | 1.64 | 2.94 | 0.78 | 0.68 | 28.66 | 38.01 | -0.10 | 0.65 |
| | $\tau_3$ | 0.75 | 0.61 | 1.77 | 3.20 | 0.76 | 0.66 | 25.31 | 37.82 | -0.67 | 1.34 |
| | $\tau_4$ | 0.66 | 0.61 | 1.09 | 3.40 | 0.71 | 0.30 | 21.77 | 85.80 | -0.31 | 3.83 |
| 1330 | $\tau_1$ | 0.35 | 0.25 | 2.02 | 2.70 | 0.71 | 0.60 | 69.78 | 78.18 | 0.37 | 0.87 |
| | $\tau_2$ | 0.76 | 0.5 | 1.54 | 3.27 | 0.81 | 0.69 | 27.56 | 40.98 | 0.23 | 0.63 |
| | $\tau_3$ | 0.77 | 0.59 | 1.66 | 3.18 | 0.80 | 0.70 | 23.16 | 34.17 | -0.46 | 0.76 |
| | $\tau_4$ | 0.84 | 0.64 | 0.98 | 2.46 | 0.76 | 0.66 | 23.30 | 43.89 | -0.56 | 1.23 |

**Figure 1.** A conceptual diagram of the methodology. On the left side is a representation of predicting daily incoming short wave radiation ($R_{Sd\_pred}$). The ANN is trained to learn the system response to a combination of explanatory variables i.e. instantaneous incoming short wave radiation ($R_{Si}$), instantaneous exo-atmospheric shortwave radiation ($R_{Si}TOA$), daily exo-atmospheric shortwave radiation ($R_{Sd}TOA$), solar zenith angle ($\theta_Z$), and day length ($L_D$), by being fed with a sample data of observed daily incoming short wave radiation ($R_{Sd\_obs}$) which is the dependant variable. On the right side are methods of upscaling instantaneous ($ET_i$) to daily $ET$ ($ET_d$) using our $R_S$–based method (a) and other two approaches (b, c) are the $R_{STOA}$ and $EF$-based methods respectively used which are used for comparison.

### $R_{Sd}$ estimation approach

**Inputs**
$R_{Si}$
$R_{Si}TOA$
$R_{Sd}TOA$
$\theta_Z$
$L_D$

**Output**
$R_{Sd\_obs}$

ANN → $R_{Sd\_pred}$

### $ET_d$ upscaling approach

$a)\ ET_{d\_pred} = ET_i \dfrac{R_{Sd}}{R_{Si}}$

$b)\ ET_{d\_pred} = ET_i \dfrac{R_{Sd}TOA}{R_{Si}TOA}$

$c)\ ET_{d\_pred} = [1.1(R_n - G)_d]\dfrac{ET_i}{(R_n - G)_i}$

$ET_{d\_pred}$

Validation against $ET_{d\_obs}$

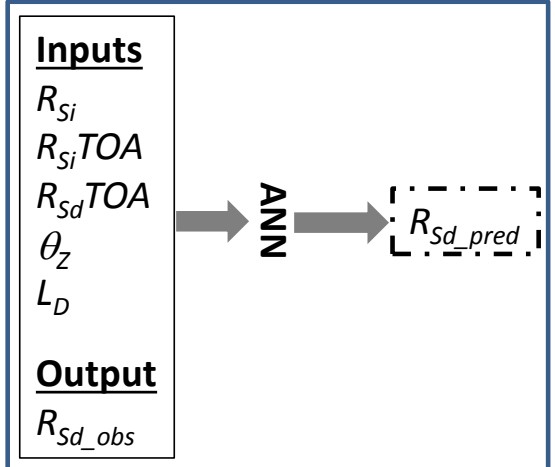

**Figure 2**. Schematic representation of a simple artificial network model. The artificial neuron has five input variables, for the intended output. These inputs are then assigned weights (*W)* and bias (b), and the sum of all these products ($\sum$) is fed to an activation function ($f$). The activation function alters the signal accordingly and passes the signal to the next neuron(s) until the output of the model is reached (Mathworks, 2015).

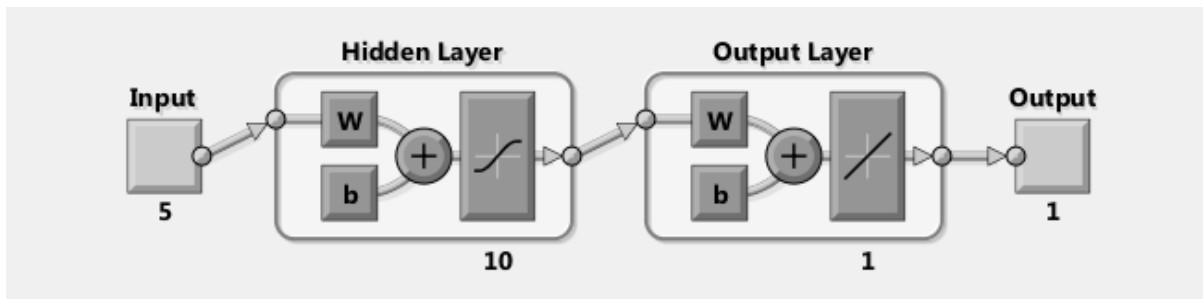

**Figure 3.** Distribution of 126 sites of the FLUXNET eddy covariance network used in the present study with 85 and 41 sites for training and validation, respectively between the years 1999 and 2006.

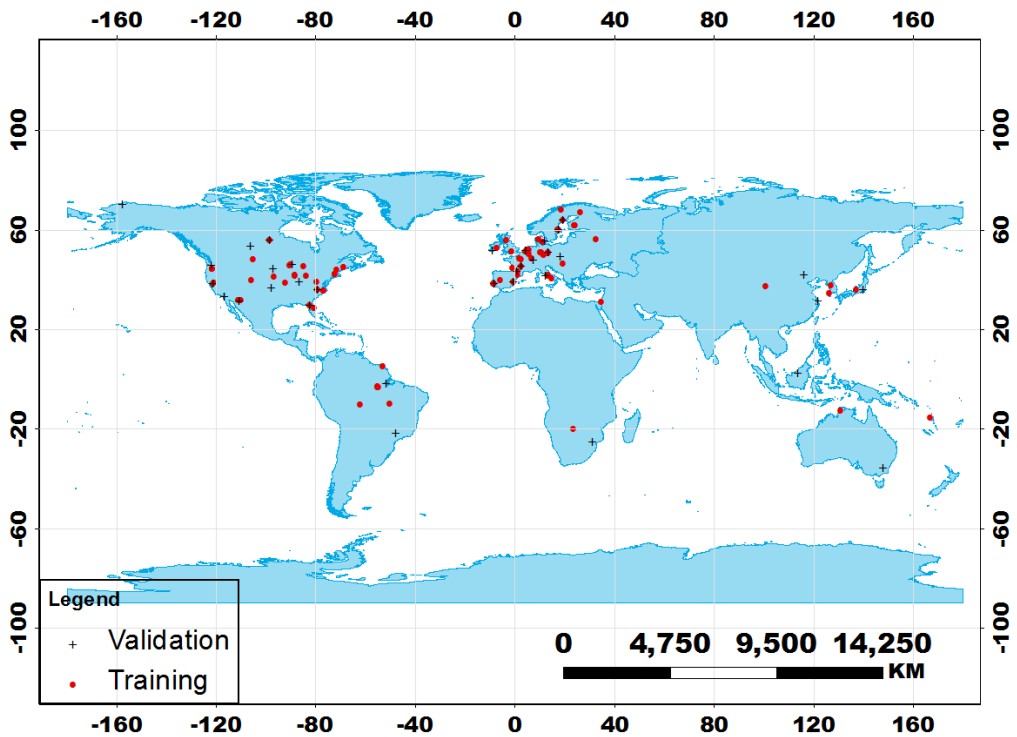


**Figure 4.** Statistical metric of $R_{Sd\_pred}$ by ANN for different time-of-day. As the study is intended for remote sensing application, we demonstrate the potential of the method for future research in the case where satellite will be used and as such we pick MODIS overpass time as an example to highlight on the predictive ability of the ANN at the specific overpass times.

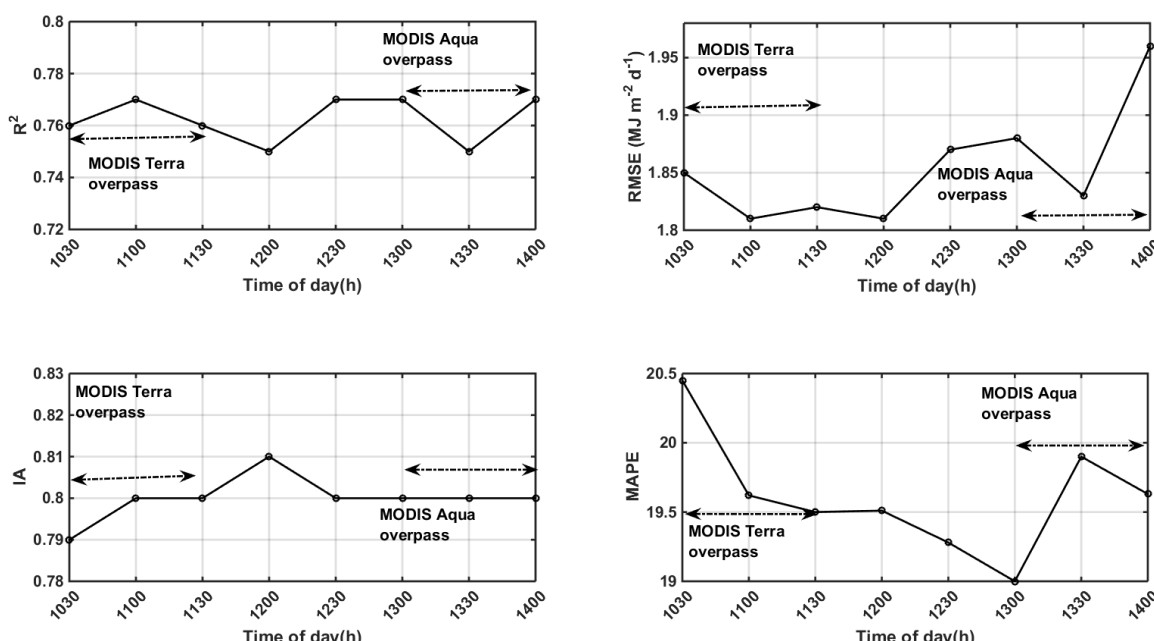

**Figure 5.** Scatter plots between $R_{Sd\_obs}$ versus $R_{Sd\_pred}$ for different levels of daily atmospheric transmissivity classes ($\tau$) from (a) 1100 and (b) 1330 hours upscaling. Here $\tau_1$–$\tau_4$ represent daily atmospheric transmissivity of four different class, $0.25 \geq \tau \geq 0$, $0.50 \geq \tau \geq 0.25$, $0.75 \geq \tau \geq 0.50$, and $1 \geq \tau \geq 0.75$, respectively, with $\tau_1$ signifying high degree of cloudiness (or overcast skies) whereas $\tau_4$ indicates clear skies.

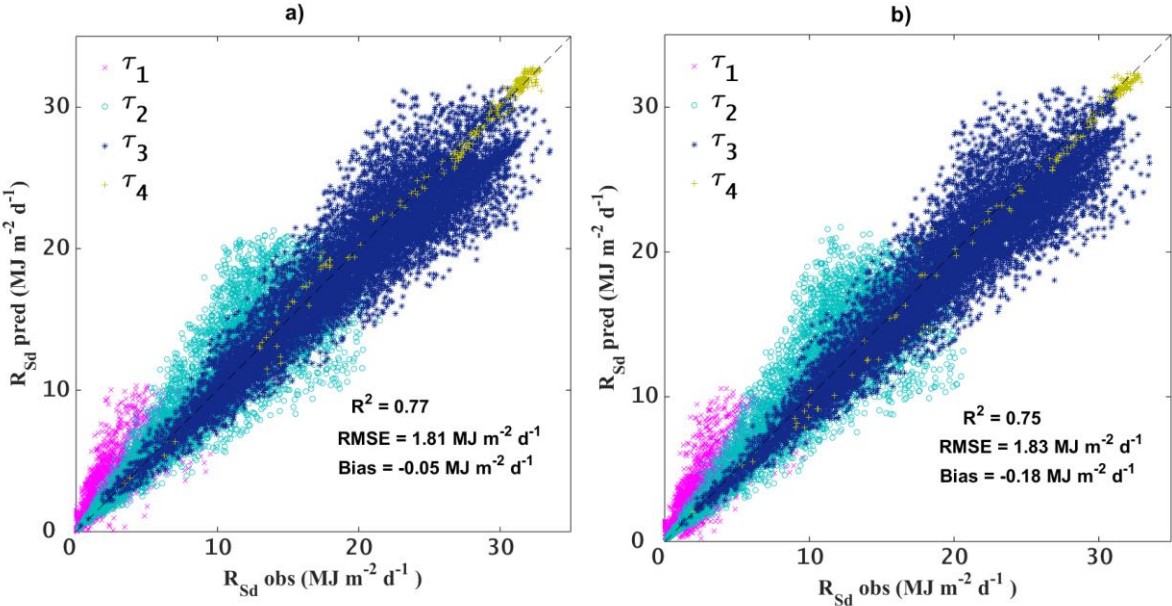

**Figure 6.** Statistical summary of $ET_{d\_pred}$ for different time-of-day using Eq. (1) based on $R_{Si}$ and $R_{Sd\_pred}$. As the study is intended for remote sensing application, we once again demonstrate the potential of the method for future research in the case where satellite will be used and as such we pick MODIS Terra-Aqua overpass time.

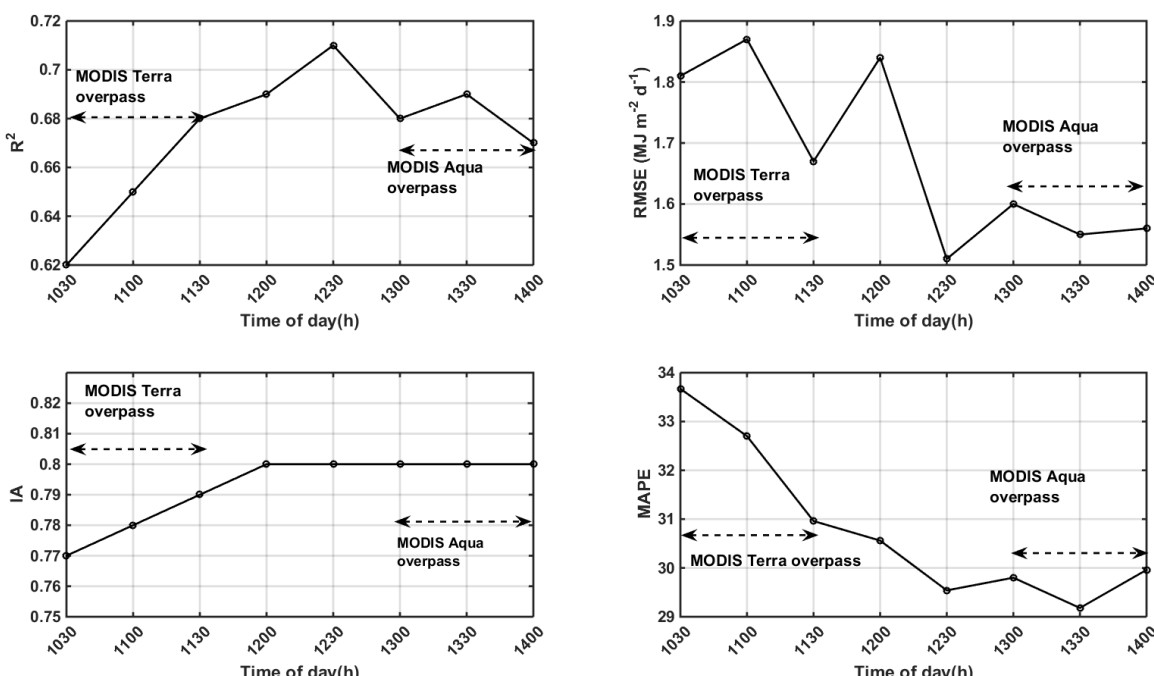

**Figure 7.** $ET_{d\_pred}$ obtained through eq. (1) versus $ET_{d\_obs}$ for different levels of τ from both forenoon (a) and afternoon (b) upscaling (1100 and 1300 h daytime hours).

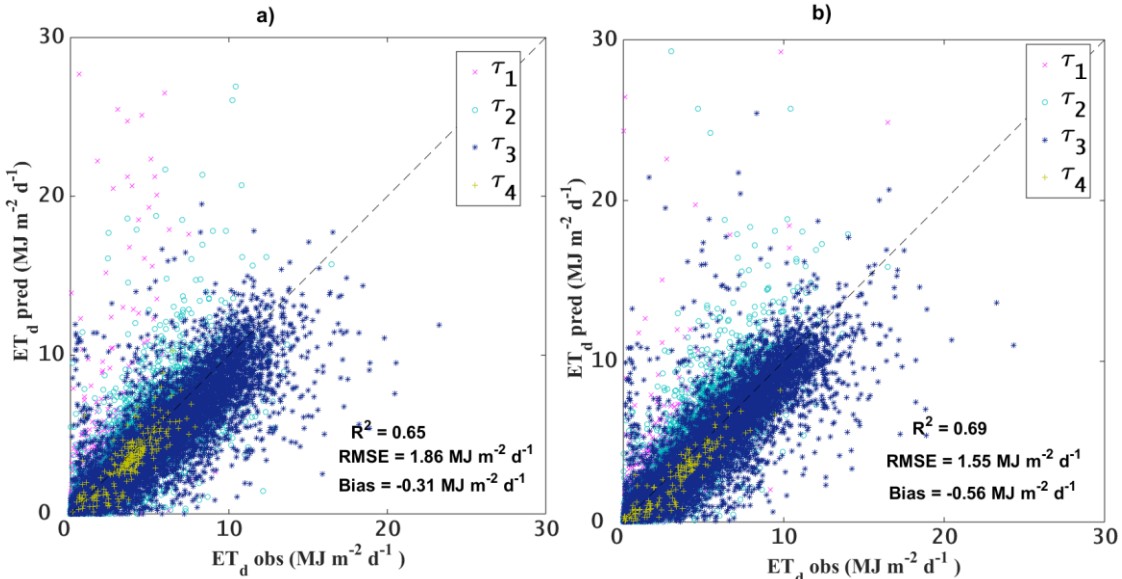

**Figure 8.** Assessing the statistical metrics of $ET_{d\_pred}$ (using eq.1) for different levels of daily atmospheric transmissivity classes (representing cloudy to clear skies) for both 1100h and 1330h time-of-day $ET_i$ scaling.

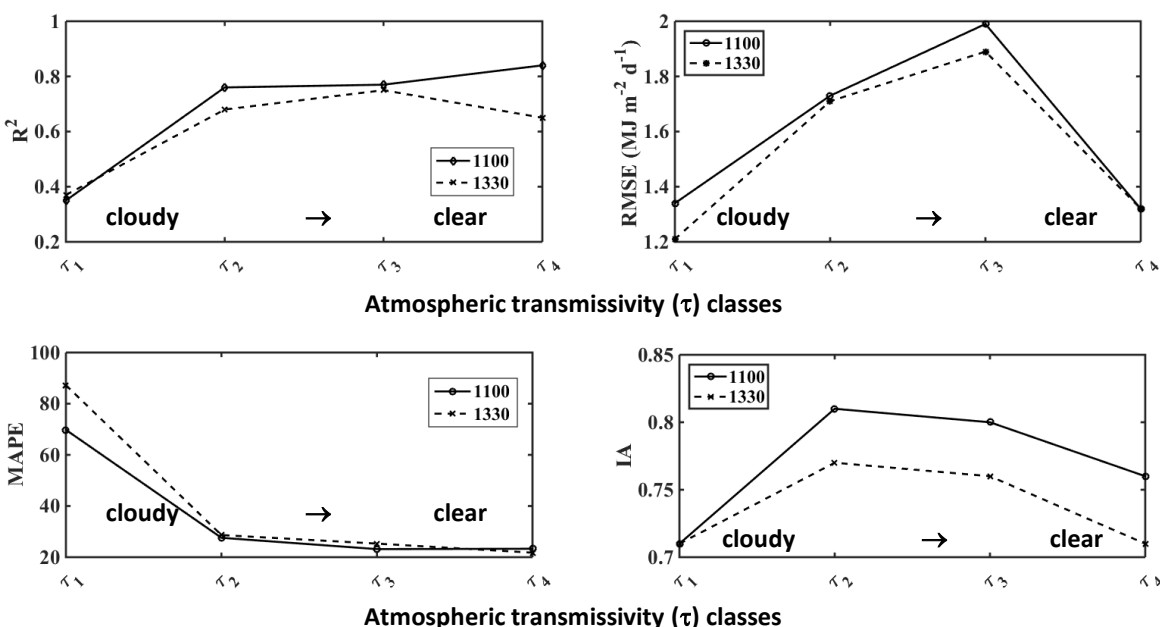

**Figure 9.** An intercomparison of $ET_{d\_pred}$ error statistics (RMSE and MAPE) for different levels of atmospheric transmissivity classes based on two different ANN training (ANN trained with shortwave radiation and astronomical variables only; and ANN trained with radiation, astronomical variables, soil moisture and rainfall) based on 1100h and 1330h time-of-day $ET_i$ scaling.

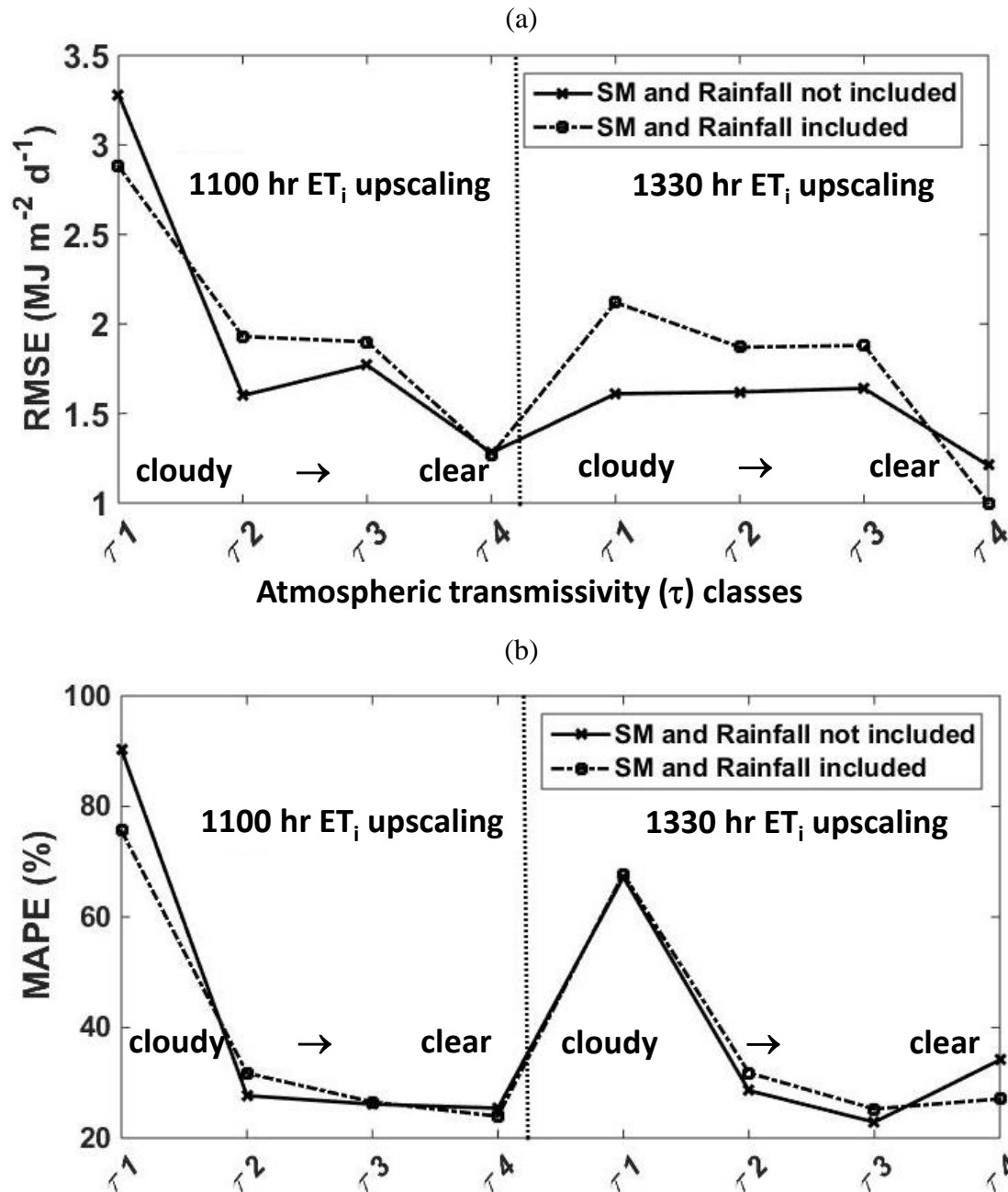

**Figure 10.** Time series comparison between observed and predicted $ET_d$ for four representative sites located in Australia, Brazil, South Africa and Sweden.

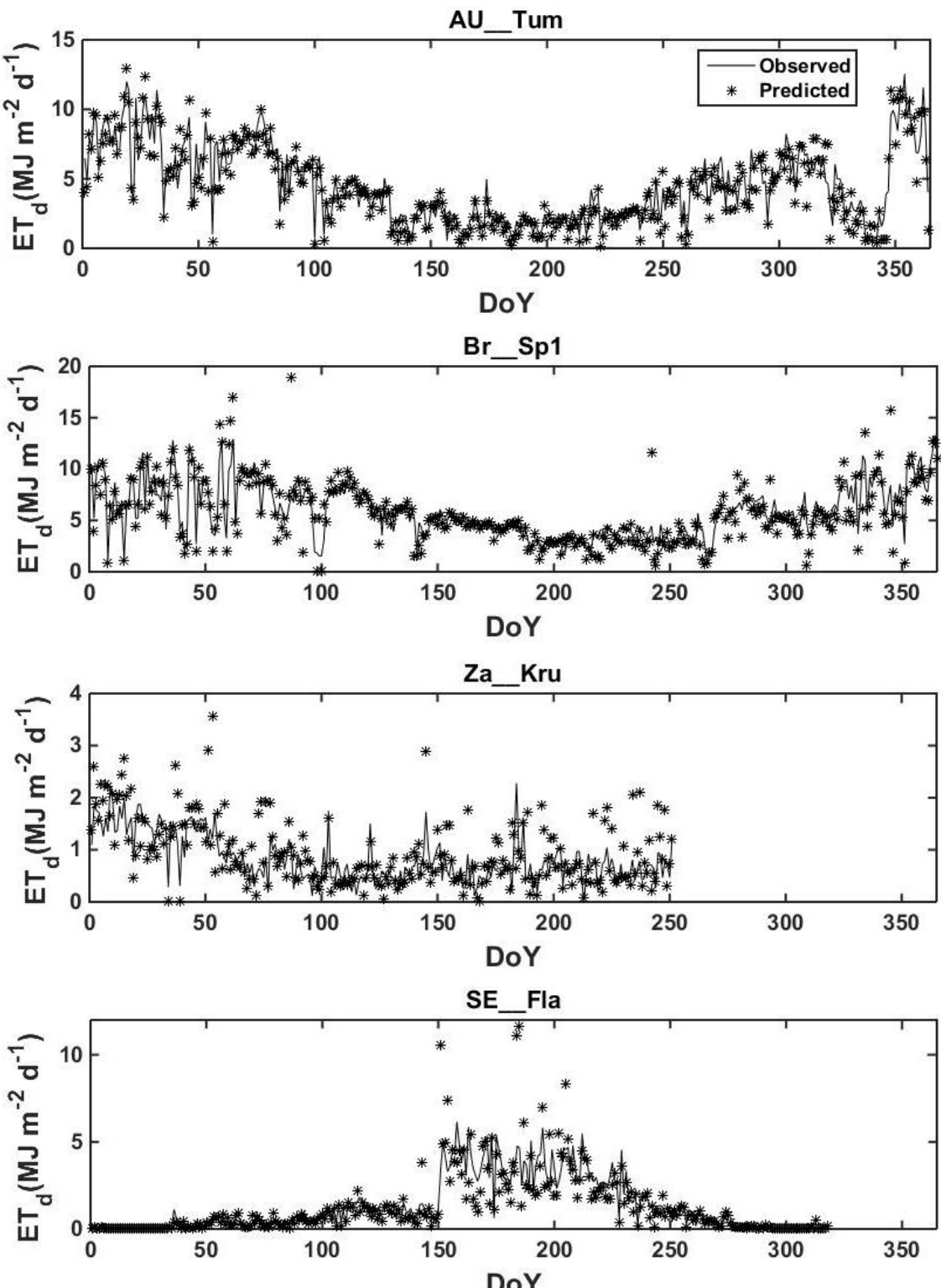

**Figure 11.** Biome specific error characteristics of $ET_{d\_pred}$ displaying the box plots of RMSE and coefficient of determination ($R^2$) from both $R_S$-based and $R_STOA$-based $ET_i$ upscaling. The biome classes are evergreen broadleaf forest (EBF), evergreen needleleaf forest (ENF), deciduous broadleaf forest (DBF), shrubland (SH), cropland (CRO), and grassland (GRA), respectively.

(a) $R_S$-based RMSE of $ET_{d\_pred}$     (b) $R_S$-based $R^2$ of $ET_{d\_pred}$

(c) $R_STOA$-based RMSE of $ET_{d\_pred}$     (d) $R_STOA$-based $R^2$ of $ET_{d\_pred}$

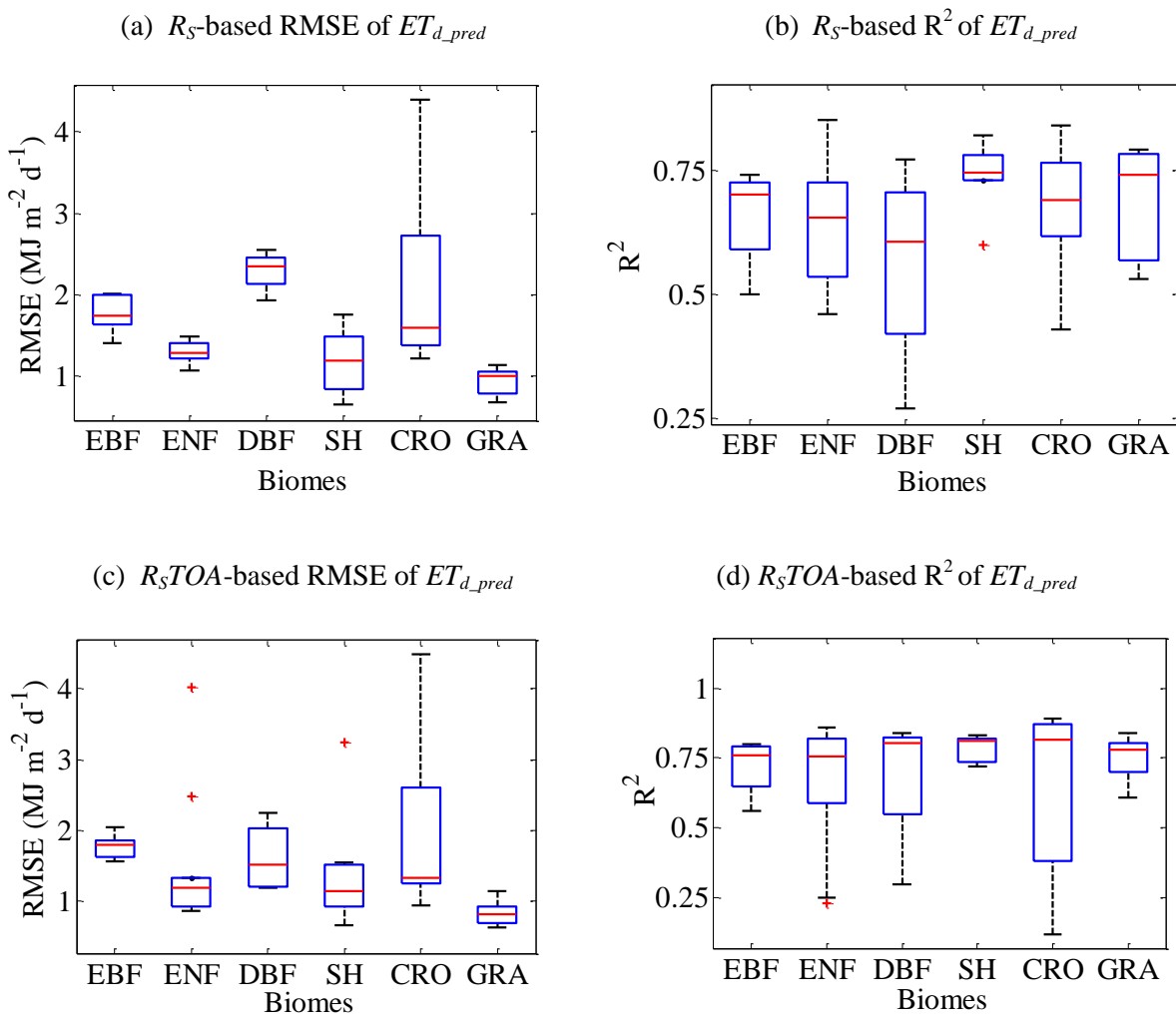


**Figure 12.** Statistical metrics of $ET_{d\_pred}$ from three different $ET_i$ upscaling approaches [shortwave incoming radiation ($R_S$), exo-atmospheric shortwave radiation ($R_STOA$) and evaporative fraction ($EF$)] at different temporal scales based on $ET_i$ measurements at (a) 1100h and (b) 1330h time-of-day.

(a) 1100 h $ET_i$ upscaling

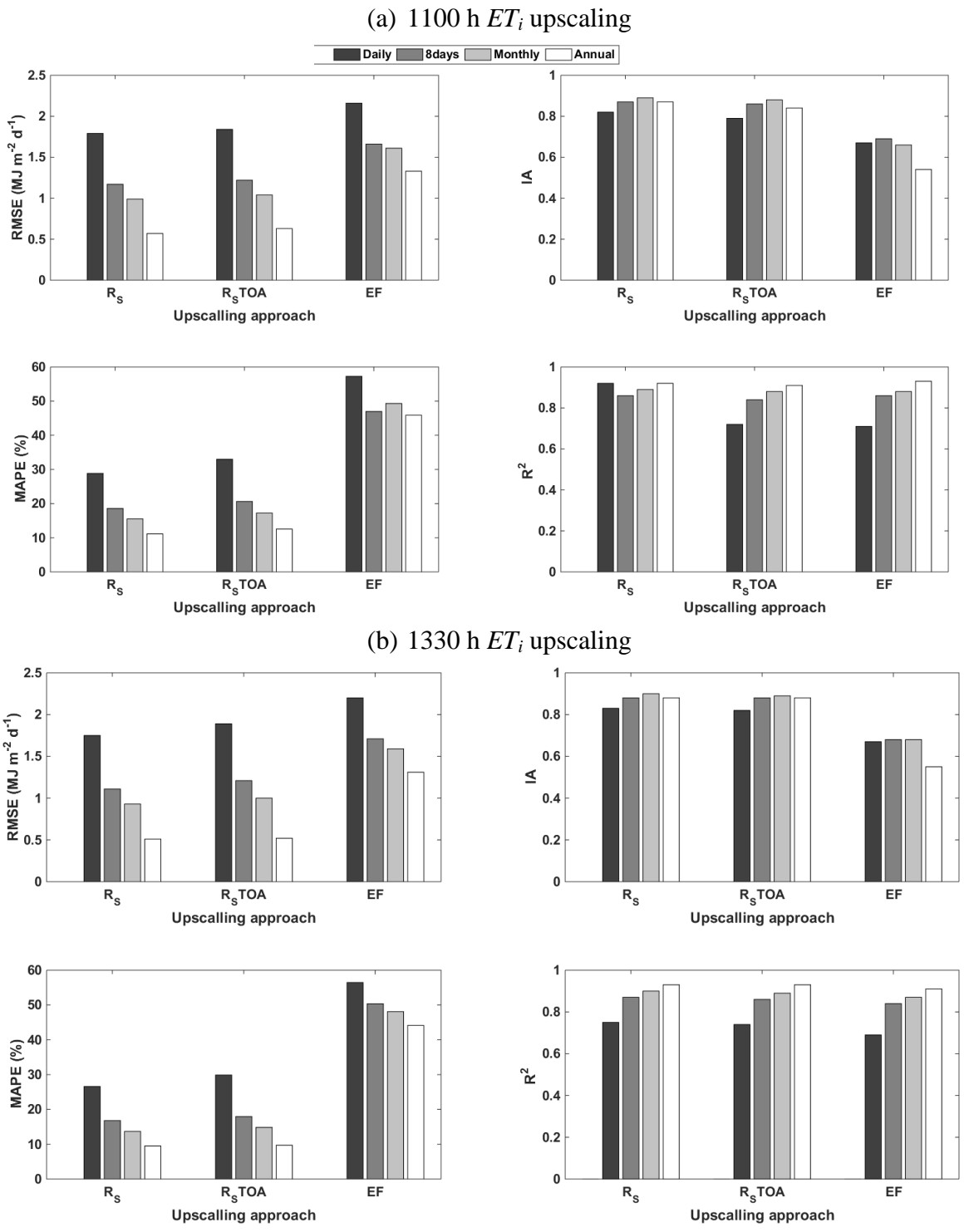

(b) 1330 h $ET_i$ upscaling

**Figure 13.** Illustrative examples of the sensitivity of $ET_{d\_pred}$ error statistics ($R^2$ and RMSE) to the different biome type scenarios of ANN training. Here, Case1 consist of training the ANN with forest (FOR) datasets and evaluating ANN predicted $ET_d$ statistics on non-forest biomes, Case2 consist of training the ANN with non-forest datasets and evaluating ANN predicted $ET_d$ statistics on forest biomes, Case3 consist of training the ANN with both forests and non-forest datasets and evaluating ANN predicted $ET_d$ statistics on all the biomes.

(a) $R^2$ of $ET_{d\_pred}$ for three different ANN training scenarios

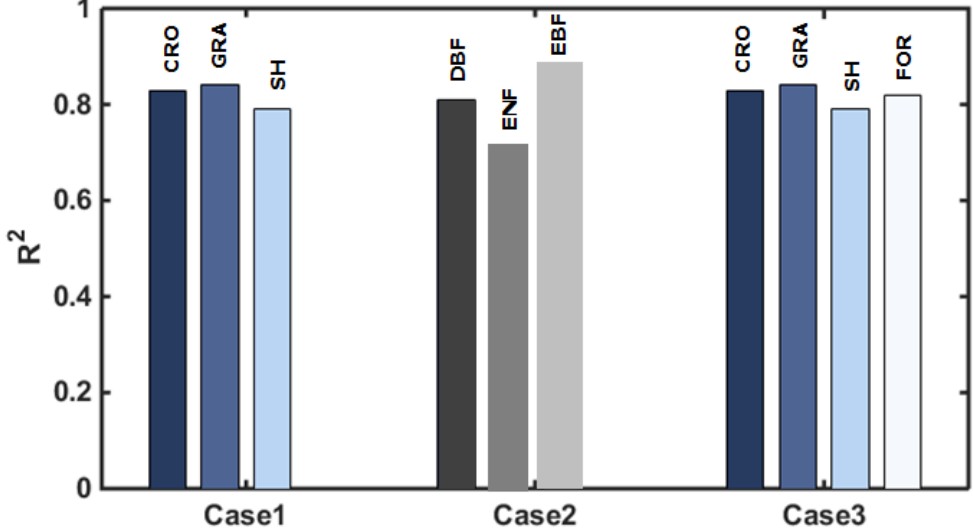

(b) RMSE of $ET_{d\_pred}$ for three different ANN training scenarios

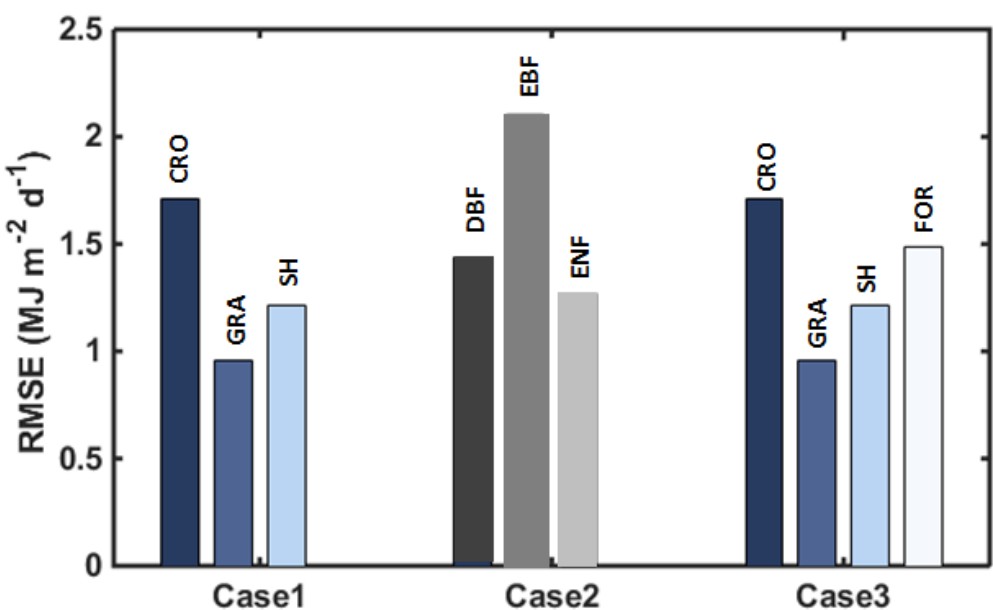

