# Peer review of "Upscaling instantaneous to daily evapotranspiration using modelled daily shortwave radiation for remote sensing applications: an Artificial Neural Network approach"

_Hydrology and Earth System Sciences, 2016_

## Referee Comment (RC1) · Anonymous Referee #1 · 17 Aug 2016

Overview:

Loise Wandera et al., presented a study that upscaled instantaneous evapotranspiration (ET) to daily ET. I enjoyed reading the manuscript. The paper was generally well written, the method is robust, and evaluation is rigorous. It's scientifically significant in terms of improving remote sensing ET product. The overall quality of this work is good, but could be further improved by considering the following comments. Below I have several major concerns, which may require additional work.

Major comments:

[Figure]

1. Energy budget closure problem at FLUXNET.

Energy budget imbalance has long been identified at FLUXNET sites. The imbalance is about -40% - +20%, indicating latent heat/sensible heat fluxes might be underestimated by up to 40%. Indeed, the energy imbalance is an existing fact we have to accept, I guess there is little can be done to overcome it in this particular study. But my concerning is: if an ANN model is trained by FLUXNET data, how much confidence do we have when we apply it to satellite retrieval? The energy budget close problem affects the results in two ways: (1) the overall robustness of the proposed upscaling method (Rs method); (2) comparison of Rs method with the evaporative fraction based upscaling (EF method Eqn. 5). However, the exo-atmospheric irradiance method is not affected (Eqn. 6). I guess the authors must be aware of this issue; it would be better to literally discuss them in the results section.

2. Cloudy-sky issue

The biggest problem of the proposed upscaling method (Rs method) is that the ANN model does not include any information about "cloudiness". Therefore, model performance under cloudy-sky condition (or low atmospheric transmissivity) is much worse than clear-sky condition. One way to tackle it, is to use climatology precipitation data. Rainfall (highly related to cloudiness) has seasonal pattern, at least for some regions (e.g., tropical rainforest, savanna). Similarly, dry season-wet seasons could provide ANN model with additional information about "possibility" of the "cloudy-sky condition" during a certain time period. In Figure 7, the overestimation of ET under cloudy sky condition is "systematic", meaning there might be a simple way to "systematicly" down-regulate the ET as long as the ANN model knows it's a cloudy day.

3. FLUXNET site selection.

It was stated that the partition of data into training and validation was randomly selected. However, it's not clear whether the selected training sites are representative or not. For example, if plot out mean annual precipitation of the all training data, does
it cover a full range of (from dry to wet) rainfall regimes? For each vegetation type, how much percentage of data is selected to train the model? FLUXNET has more forest sites than grass/shrub sites. Are grass/shrub sites less represented in the training dataset?

Following question: is the ANN model sensitive the FLUXNET site selection? This could be evaluated by doing e.g., 10 ensemble of random selection of FLUXNET sites. And check the difference among the resultant 10 ANN models?

4. Crop ET

I think the proposed method might be only suitable for estimating natural terrestrial ecosystem ET. There is large bias of crop ET estimation (Figure 9). That could be due to irrigation? Land management? Those anthropogenic factors (largely alter land surface water budget) is not included in the ANN model and the ET estimation.

5. Vegetation control on ET

The proposed upscaling method is based on the idea that higher available energy (Rs) lead to higher evaportranspiration (ET) (Eqn. 1). It basically assumes that the Bowen ratio does not change during the daytime, so that instantaneous ET/Rs is equal to daily ET/Rs. However, it ignores the important fact that ET is also mediated by vegetation via stomata control. For example, trees and grass have dramatically different stamata density, stomata size. Therefore, their stomata open/closure and its control on water vapor conductance are different. The question is: it is worthwhile to add biome type information in the ANN model? Is it possible to further improve the results (Figure 9) for forest sites by considering biome type information in the ANN model and ET estimates?

Minor comments:

Page2

L4. a key challenge in mapping regional ET using polar orbiting sensors

L6. On the terrestrial surface -> remove

L8. The approach relies on . . . -> remove

L16 derived from simple mathematical computation -> replace: e.g., solar zenith angle, day length

L20. Based on the measurements from 126 sites -> remove

L20. Rs-based upscaling produced . . .

Page3

L7, Et variability is influenced by (1) available energy received, (2) soil moisture supply and (3) vegetation mediation. I think the third one is missing here. To be complete, the three key factors should all be fairly discussed in the introduction.

L9. "Therefore" is not appropriate here, there is no cause-effect relationship here. Better start a new paragraph and discuss the major challenges in Et upscaling.

L19. Estimate Rsd form any specific time-of-day Rsi information. But isn't the value of this study is to predict Rsd based at satellite local crossing time (e.g., 10:30, 13:30)?

L22. In order -> remove

L24. ANN is a non-linear model .... Multi-layer perceptron (MLP) is . . .. These sentences belongs to method section.

Page 5.

L13. Cloudiness is a phenomenon . . .. These sentences belong to discussion section.

Page 6.

L6. Two question: (1) Does Equn. 1 assume the Bowen ratio is constant during day-time? (2) Does it ignore the night time ET, which could be large when surface wind

speed is high?

Page 8.

L16. In a percentage ratio of 80:15:15. Is this right? Shouldn't be 80:15:5 or 70:15:15?

Page 10.

L9. We first evaluate the efficacy of the ANN method for predicting Rsd.

L12. As obtained following the methodology described in the section 2.1 -> remove

L13. Showing -> including

L14. From the analysis it is apparent that -> remove

Page 11.

L1. Figure 5 evaluates the Rsd_pred under different level of clear sky transmissivity ($\tau$).

L3. What if the ANN model includes "clear sky transmissivity ($\tau$)", would model performance under cloudy sky condition be improved?

L16. Using Rsd_pred/Rsi as a scaling factor following eq. 1 -> remove

Page 12.

L1. Figure 7 compares ETd_pred against ETd_obs for different level of daily $\tau$. The overall RMSE, MAPE . . .

L4. Given that the overestimation is a systematic, is it possible to eliminate it or reduce it? The overestimation was due to the fact that during the specific time slot of interest (e.g., 11:30) the sky is clear while the sky is cloudy during other times. However, there could be another opposite case that sky is cloudy at e.g., 11:30 but clear at other times. It will probably lead to an underestimation of RSd_pred, and consequently underestimation of ETd_pred. I am wondering why the latter is not the case at least in

Figure 7.

L14. ... higher errors in ETd_pred can be expected. Is there a way to overcome this problem?

L24. Again, biome specific results are related to the clear-sky issue. Tropical evergreen broadleaf forests have high ET, water tends to re-cycle locally and generate rainfall. It's reasonable to see that cloudy sky condition is more frequent at tropical evergreen broadleaf forest than e.g., at grass land.

L27. ET estimations at cropland were much worse than grass. It that because e.g., irrigation? Land management? Or any other anthropogenic factors that are not considered in the ANN model? Page 13.

L20. Based on Table 2, Figure 11, RsTOA method seems successful. Under clear sky condition, it was even better than the proposed Rs method. Further, over longer time scale (annually), there is no big difference between RsTOA and Rs.

Page 16.

L1. Briefly define what is RsTOA-based metod, what is Rs method.

L4. ETd_pred are defined early in the manuscript, consider the summary as a independent section. Better not to use these acronyms, or re-define it.

L21-25. This paragraph belongs to results & discussion section.

---

## Referee Comment (RC2) · Anonymous Referee #2 · 12 Sep 2016

This paper developed and evaluated a temporal upscaling method for estimating ETd from different time-of-daytime instantaneous ET (ETi) measurements with the assumption that the ratio between daytime to instantaneous RS (RSd/RSi) is the predominant factor governing ETd/ETi ratio. However, since RSd is not measurable from the polar orbiting satellites, they first developed a robust ANN based method to upscale RSi to RSd followed by using the ratio of RSd/RSi to further upscale ETi to ETd.

Given the significance of ETd in remote sensing based water resource management from polar orbiting satellites, the overarching goal of this study is to provide an oper-

ational and robust ETi upscaling protocol for estimating ETd from any polar orbiting satellite.

I found the research idea and the methodology behind it to be very interesting. However, I have some minor comments and questions as detailed below:

1- How do you pick the training sites? Will the vegetation type and climate type (seasonal climate) have any effect on your trained ANN algorithm? Given that Fluxnet sites at least in N. America are mostly forest sites, will that have any potential impact on your trained ANN?

2- I think a paragraph on Rs and factors affecting Rs is missing from the paper. This is necessary to justify your choice of inputs for your ANN.

3- Please include discussion on why the method performs poorly over cropland (Figure 9).

4- As discussed in lines 25-27, Rsd and cloudiness are directly related. ANN has no input related to cloudiness. However, you argue that you assess the performance of ANN under cloudy sky condition based on simple cloudiness index. Please elaborate on this and include discussion in the paper. Can you use Precipitation or the index of cloudiness as an input to your ANN?

5- Since vegetation plays an important role in Evapotranspiration, it would be interesting to compare different scaling methods against the type of vegetation as well (in graphs or figures)

---

## Referee Comment (RC3) · Anonymous Referee #3 · 16 Sep 2016

This paper assesses the performance in retrieving daily solar incoming radiation from instantaneous estimates using an ANN algorithm and ancillary earth-sun geometrical parameters, and subsequently the performance of upscaling instantaneous evapotranspiration estimates to daily totals using daily solar radiation derived from the ANN as an upscaling support variable. The latter is also compared to two classically used methods (using respectively the TOA solar radiation as a support variable and the evaporative fraction selfpreservation).

Main concern:

I don't see the point of upscaling ETi to ETd for days where instantaneous observations in the optical domain are not available from satellite platforms: instantaneous ETi estimates are usually produced with instantaneous data in the optical domain, typically Thermal Infra Red data, and are therefore not computed for low transmissivities, airborne platforms excepted. Days with low instantaneous (10AM, 1:30PM) transmissivities should be left out of the study i.e. the study should restrict to clear sky conditions from either MODIS cloud mask or, better, geostationary information (the CERES algorithm mentioned here).

I therefore doubt that there is any use of the method for "Remote sensing applications" as mentionned in the title, except for UAV applications. Actually, it is interesting to note that even for clearsky conditions the ANN method shows worse performances than the classical method based on the sole earth-sun geometrical parameters.

Estimating ETR between 2 successive clearsky days is an interpolation problem (which could be also treated using ANN) which needs to be tackled also.

Main comments:

- I also share the main concern with referee 1 about Energy Balance Closure: lack of EBC should not be overlooked and is simple to correct for FLUXNET sites; it could explain the poor performance of the Evaporative Fraction method. Disregarding EBC is a major methodological flaw of the paper.

- As criticized also by referee 1, crops and semi-arid or even dry subhumid sites are underrepresented in the FLUXNET database, this should be more carefully commented. It adds up to my concern above about the practical application of the method: TIR-based daily ETR computation algorithms are particularly needed for water use monitoring in water depleted environments, much less for natural vegetation in temperate climates.

- Are the validation and the training datasets from different years ? It seems to me that

this is a requirement to use the method for future applications.

- What is the true added value of the ANN for future operational applications of the upscaling algorithm, say for an operational satellite product ? This aspect, although the original motivation of the paper, is somewhat overlooked in the discussion section. From Table 2, it appears that the TOA solar radiation-based method shows the best performances.

- For cloudy conditions the ETR upscaling method using instantaneous solar radiation as part of the training (even from another site) performs slightly better than that based on the sole TOA solar radiation : is it mostly due to the fact that the ANN adds information on actual incoming radiation obtained at a "nearby" FLUXNET location ?

Minor comments:

In introduction one should add a review of which upscaling support variables can be derived from remote sensing data directly, which can be obtained indirectly from either RS data or any other distributed routinely produced data and those not obtainable from remote sensing or other distributed operational datasets.

How do you manage nighttime conditions ?

Move P5L1-4 to the end of this section and precise the variables fed by ANN upfront there.

P8 L11-15: It is not clear, why is there a testing dataset and a separate validation dataset within the training dataset ?

P9L5: Why use transmissivity rather than the ration between actual and theoretical clearsky radiations to separate the various cloudiness bins ? (in order at least to separate winter conditions with lower clear sky transmissivity from summer conditions)

P14L10: "would likely": this can be checked, is it the case ?

P13L12: "reasonable" > "reasonably"

---

## Author Comment (AC1) · 13 Oct 2016

**Reviewer 1 (R1):**

**We would like to thank R1 for the detailed comments.**

1. **Energy budget closure problem at FLUXNET**. Energy budget imbalance has long been identified at FLUXNET sites. The imbalance is about -40% - +20%, indicating latent heat/sensible heat fluxes might be underestimated by up to 40%. Indeed, the energy imbalance is an existing fact we have to accept, I guess there is little can be done to overcome it in this particular study.

   **Response:** Good point. We propose to include an intercomparison of $ET_i$ upscaling results including both energy balance closure and non-closure in the revised version of the manuscript.

2. But my concerning is: if an ANN model is trained by FLUXNET data, how much confidence do we have when we apply it to satellite retrieval? The energy budget close problem affects the results in two ways: (1) the overall robustness of the proposed upscaling method (Rs method); (2) comparison of Rs method with the evaporative fraction based upscaling (EF method Eqn. 5). However, the exo-atmospheric irradiance method is not affected (Eqn. 6). I guess the authors must be aware of this issue; it would be better to literally discuss them in the results section.

   **Response:** Regarding R1's concern on the impact of surface energy balance closure on the performance of $ET_d$ evaluation, **it is important to mention that the implicit assumption in remote sensing based $ET_i$ retrieval is the closure of surface energy balance**. Therefore, for the remote sensing retrievals, the energy balance closure problems will not affect the $ET_d$ estimates in the current framework of ANN. **However, for the validation of remote sensing based $ET_d$ retrievals, surface energy balance fluxes from eddy covariance measurements need to be closed.**

   In the present study, the closure problem of surface energy balance will affect the evaluation statistics of all the three methods, and therefore, we propose to include an intercomparison of $ET_i$ upscaling results including both energy balance closure and non-closure in the revised version. **As compared to the EF and $R_S$TOA approach, the Rs method is more robust with regards to *ET* scaling on a daily time frame since the method carries maximum information on the cloudiness, which is a key limiting factor in upscaling of $ET_i$ to $ET_d$.**

With reference to Eq. (1), the network developed is intended to develop an operational method to directly upscale $ET_i$ (estimated from polar orbiting satellites) to $ET_d$ based on the ratio of daily to instantaneous shortwave radiation ($R_{Sd}$ and $R_{Si}$). Given there is no direct method to directly estimate $R_{Sd}$ from remote sensing satellite, we trained an ANN with the FLUXNET observations of $R_{Si}$ and $R_{Sd}$, and validated the model to predict $R_{Sd}$ over independent sites, followed by using $R_{Sd}/R_{Si}$ ratio to convert $ET_i$ to $ET_d$. The datasets used for the ANN development covers a wide range of biome, climate, and variable sky conditions. Therefore, we assume the $R_{Sd}$ prediction from ANN to capture a broad spectrum of radiative forcing, which is also reflected in the independent validation of RSd and $ET_d$ (Fig. 5, Fig. 7, Table 2). The performance of this model for satellite retrieval of $R_{Sd}$ (from $R_{Si}$) is dependent on the accuracy of $R_{Si}$ retrieval(Loew, Peng, & Borsche, 2016). We shall make this point explicit in the conclusion section. Also, the distribution of sites over the tropics, Africa, and SE Asia are poor, and more sites over these regions are expected to make the ANN model more robust, which will also be mentioned in the revised manuscript.

3. **Cloudy-sky issue**. The biggest problem of the proposed upscaling method (Rs method) is that the ANN model does not include any information about "cloudiness". Therefore, model performance under cloudy-sky condition (or low atmospheric transmissivity) is much worse than clear-sky condition. One way to tackle it, is to use climatology precipitation data. Rainfall (highly related to cloudiness) has seasonal pattern, at least for some regions (e.g., tropical rainforest, savanna). Similarly, dry season-wet seasons could provide ANN model with additional information about "possibility" of the "cloudy-sky condition" during a certain time period. In Figure 7, the overestimation of ET under cloudy sky condition is "systematic", meaning there might be a simple way to "systematically" down regulate the ET as long as the ANN model knows it's a cloudy day.

**Response**: Including cloudiness as an input variable of the network during training process would significantly enhance the performance of the network. Use of daily precipitation as an indicator of cloudiness would have been the most appropriate approach in this circumstance. However the cloud information available from alternative sources  e.g. from the Clouds and Earth's Radiant Energy System (CERES), the International Satellite Cloud Climatology Project–Flux Data (ISCCP-FD), and Global Energy and Water cycle Experiment Surface Radiation Budget (GEWEX-SRB) are available at coarse spatial resolution and there will be a scale mismatch. However, the precipitation data was not consistently available for most of the sites and the data gaps were significant to alter the sampling sizes. However for future

studies, including cloudiness or daily precipitation as a variable in the training of the ANN to predict $R_{Sd}$ is highly recommended. On the issue of systematic errors as a result of cloud conditions, we certainly expect overestimation or underestimation.

4. **FLUXNET site selection**. It was stated that the partition of data into training and validation was randomly selected. However, it's not clear whether the selected training sites are represent it cover a full range of (from dry to wet) rainfall regimes? For each vegetation type, how much percentage of data is selected to train the model? FLUXNET has more forest sites than grass/shrub sites. Are grass/shrub sites less represented in the training dataset? Following question: is the ANN model sensitive the FLUXNET site selection? This could be evaluated by doing e.g., 10 ensemble of random selection of FLUXNET sites. And check the difference among the resultant 10 ANN models?

**Response**: Since this analysis was based on FLUXNET sites distributed across 0-90 degrees latitude north and south, the training datasets covers substantial climatic and vegetation variability. The percentage distribution of the training data according to vegetation type was; 23% crops, 31% deciduous broadleaf forest, 10% evergreen broadleaf forest, 20% evergreen need leaf forest, 8% grassland, 7% shrubs and 1% aquatic as indicated in table S1. The number of grassland and shrubs as indicated were relatively less as compared to the crops and forests sites. **However, biome specific error statistics (Fig. 9) indicted the absence of any systematic errors due to vegetation sampling with the exception of EBF. Availability of more EBF sites in the training datasets is expected to reduce the cloudy sky errors substantially. We shall elaborate this discussion in the revised manuscript.**

5. **Crop ET.** I think the proposed method might be only suitable for estimating natural terrestrial ecosystem *ET*. There is large bias of crop *ET* estimation (Figure 9). That could be due to irrigation? Land management? Those anthropogenic factors (largely alter land surface water budget) is not included in the ANN model and the *ET* estimation.

**Response:** Yes, in the current framework the approach would be best suited for natural ecosystem. However, inclusion of daily soil moisture and rainfall in the ANN might improve the $ET_d$ prediction in irrigated agro ecosystems. Given the rainfall and soil moisture measurements are not available in all the sites, we propose to use a subset of sites to test this hypothesis where rainfall and soil moisture information are available.

Further having many explanatory variables (e.g., land management, irrigation statistics, anthropogenic factors) to train the ANN, we risk overfiting the model and hence introducing bias.

6. **Vegetation control on ET.** The proposed upscaling method is based on the idea that higher available energy (Rs) lead to higher evapotranspiration (*ET*) (Eqn. 1). It basically assumes that the Bowen ratio does not change during the daytime, so that instantaneous ET/Rs is equal to daily ET/Rs. However, it ignores the important fact that *ET* is also mediated by vegetation via stomata control. For example, trees and grass have dramatically different stomata density, stomata size. Therefore, their stomata open/closure and its control on water vapor conductance are different. The question is: it is worthwhile to add biome type information in the ANN model? Is it possible to further improve the results (Figure 9) for forest sites by considering biome type information in the ANN model and *ET* estimates?

**Response:** This is indeed a very good point and needs to be explicitly discussed in the manuscript. The stomatal and biophysical constraints are generally imposed in satellite based ETi retrieval schemes. However the carry over effects of the stomatal control on daily *ET* is indeed overlooked. We assume the inclusion of daily soil moisture and rainfall in the ANN framework will implicitly include the stomatal control at the daily time scale. The additional analysis proposed in the previous response would be helpful in this context. Therefore, instead of biome type information, we would rely on the daily soil moisture and rainfall for a subset of sites, and include a comparative analysis of the current ANN framework (without soil moisture and rainfall) with a modified ANN framework (including soil moisture and rainfall). The new results will also be explicitly discussed in the revised version of the manuscript.

**Minor comments**

**Page2**

7. L4. a key challenge in mapping regional *ET* using polar orbiting sensors

   **Response:** Necessary changes will be incorporated.

8. L6. On the terrestrial surface -> remove

   **Response:** Necessary changes will be incorporated.

9. L8. The approach relies on : : : -> remove

    **Response:** Necessary corrections will be made.

10. L16. derived from simple mathematical computation -> replace: e.g., solar zenith angle, day length

    **Response:** Changes will be made as suggested.

11. L20. Based on the measurements from 126 sites -> remove

    **Response:** Will be removed.

12. L20. Rs-based upscaling produced

    **Response:** Necessary changes will be incorporated

**Page3**

13. L7. ET variability is influenced by (1) available energy received, (2) soil moisture supply and (3) vegetation mediation. I think the third one is missing here. To be complete, the three key factors should all be fairly discussed in the introduction

    **Response:** Good point. We shall include the vegetation controls on ET in the introduction.

14. L9. "Therefore" is not appropriate here, there is no cause-effect relationship here. Better start a new paragraph and discuss the major challenges in Et upscaling

    **Response:** Agreed.

**Page4**

15. L19. Estimate $R_{sd}$ form any specific time-of-day $R_{SI}$ information. But isn't the value of this study is to predict $R_{sd}$ based at satellite local crossing time (e.g., 10:30, 13:30)?

    **Response:** The aim of this study is to help develop an approach that would help in the upscaling of $ET_i$ (retrieved at satellite overpass time) to $ET_d$. The value of this study consists of exploiting $R_{Si}$ information at satellite local crossing time to predict $R_{Sd}$ which is not directly retrievable from any polar orbiting satellites, so that the ratio of $R_{Sd}/R_{Si}$ can be further used to upscale $ET_i$ to obtain daily *ET* ($ET_d$) estimates (in the framework of eqn. 1). Currently we are limited to demonstrating with MODIS overpass times (Terra and Aqua),

however in case there are new missions in the future with different local overpass time, the method would still be applicable. We shall make this description explicit in the revised manuscript.

16. L22. L22. In order -> remove

    **Response:** Will do.

17. L24. ANN is a non-linear model. Multi-layer perceptron (MLP) is.. These sentences belong to method section.

    **Response:** Necessary corrections will be made in the revised version.

**Page5**

18. L13. Cloudiness is a phenomenon. These sentences belong to discussion section.

    **Response:** Necessary changes will be incorporated in the revised version.

**Page6**

19. L6. Two question: (1) Does Eqn. 1 assume the Bowen ratio is constant during daytime? (2) Does it ignore the night time *ET*, which could be large when surface wind speed is high?

    **Response:** (1) There is no assumption of the conservation of Bowen ratio or evaporative fraction. According to eqn. 1,

    $$ET_d/ET_i \approx R_{Sd}/R_{Si}$$

    **and**

    $$ET_d/ET_i = EF_d(R_N - G)_d/EF_i(R_N\text{-}G)_i$$

    Where EF is the evaporative fraction, $R_N$ is net radiation, and G is ground heat flux.

    Therefore, eqn. 1 is based on the assumption that shortwave radiation is the principal driver of evaporative flux. Although *ET* can be limited due to both radiation and water, but in the water limited ecosystems the magnitude of $ET_i$ will also be low due to low soil moisture availability and therefore and upscaling $ET_i$ to $ET_d$ in the framework of eqn. 1 may not introduce significant error. The evidence is already seen in Fig. 9 where shrublands showed

relatively lower RMSE (despite being water limited) as compared to the forests. We shall extend this discussion in the revised manuscript.

(2) The analysis is based on 24-hour period, meaning night time *ET* contribution is implicitly considered. However, studies have ready shown that the nighttime ET in semi-arid regions contributes only 2 – 5% of the total season *ET* (Malek, 1992; Tolk, J, Howell, & Evett, 2006), and therefore does not appear to be significant.

**Page8**

20. L16. In a percentage ratio of 80:15:15. Is this right? Shouldn't be 80:15:5 or 70:15:15?

**Response:** The ratio should be 80:15:5, corrections will be made in the revised manuscript.

**Page10**

21. L9. We first evaluate the efficacy of the ANN method for predicting $R_{sd}$.

**Response:** Necessary changes will be incorporated.

22. L12. As obtained following the methodology described in the section 2.1 -> remove

**Response:** Necessary changes will be incorporated

23. L13. Showing -> including

**Response:** Necessary changes will be incorporated

24. L14. From the analysis it is apparent that -> remove

**Response:** Will be removed as suggested.

**Page 11**

25. L1. Figure 5 evaluates the $R_{sd\_pred}$ under different level of clear sky transmissivity

**Response:** Necessary changes will be incorporated in the revised manuscript.

26. L3. What if the ANN model includes "clear sky transmissivity, would model performance under cloudy sky condition be improved?

**Response:** We do not think so, because including clear sky transmissivity could make the modeling framework biased towards clear sky cases only.

27. L16. Using $R_{sd\_pred}/R_{si}$ as a scaling factor following eq. 1 -> remove

**Response:** Necessary changes will be incorporated

28. L1. Figure 7 compares $ET_{d\_pred}$ against $ET_{d\_obs}$ for different level of daily. The overall RMSE, MAPE

**Response:** Necessary changes will be incorporated.

29. L4. Given that the overestimation is a systematic, is it possible to eliminate it or reduce it? The overestimation was due to the fact that during the specific time slot of interest (e.g., 11:30) the sky is clear while the sky is cloudy during other times. However, there could be another opposite case that sky is cloudy at e.g., 11:30 but clear at other times. It will probably lead to an underestimation of $R_{Sd\_pred}$, and consequently underestimation of $ET_{d\_pred}$. I am wondering why the latter is not the case at least in Figure 7.

**Response:** This is a very good argument. With the current framework of ANN, this systematic overestimation cannot be eliminated. However, with the inclusion of daily rainfall and soil moisture in the ANN model, such overestimation tendency could be reduced.

Regarding R1's argument on finding underestimation of $ET_d$ from 1130 hr cloudy sky $ET_i$ upscaling in a predominant clear day, such cases were also found in $\tau_3$ category (Fig. 7) where clouds of data points clearly falling significantly below the 1:1 line, thus showing substantial underestimation of $ET_d$. We shall include this discussion in the revised manuscript.

30. L14. higher errors in $ET_{d\_pred}$ can be expected. Is there a way to overcome this problem?

**Response:** One of the probable ways to overcome the errors in cloudy sky is to incorporate daily rainfall and soil moisture in the ANN. This argument will be made explicit in the revised manuscript.

31. L24. Again, biome specific results are related to the clear-sky issue. Tropical evergreen broadleaf forests have high ET, water tends to re-cycle locally and generate rainfall. It's reasonable to see that cloudy sky condition is more frequent at tropical evergreen broadleaf forest than e.g., at grass land.

32. L27. ET estimations at cropland were much worse than grass. It that because e.g., irrigation? Land management? Or any other anthropogenic factors that are not considered in the ANN model? Page 13.

    Response: Yes, the farm management practice especially irrigation might have impact on the output for example in a case where irrigation was carried out for three consecutive days yet the sky conditions were consistently cloudy would present a challenge. We shall explicitly mention this in the discussion section of manuscript.

33. L20. Based on Table 2, Figure 11, $R_S$TOA method seems successful. Under clear sky condition, it was even better than the proposed Rs method. Further, over longer time scale (annually), there is no big difference between $R_S$TOA and Rs.L20:

    Response: Agreed and discussed also in the manuscript. As shown in Table 2, relatively lower RMSE of RsTOA for atmospheric transmissivity class above 0.75 reveals that under pristine clear sky conditions RsTOA can be successfully used to upscale $ET_i$. However, one of the main reasons for the differences in RMSE between Rs and RsTOA method for daily transmissivity above 0.75 could be due to the fact that if $ET_i$ upscaling is performed from a cloudy instance for a predominantly clear sky day, then such RMSE difference between the two different upscaling methods is expected. These results also showed the probability of a hybrid $ET_i$ upscaling method by combining Rs-method (for transmissivity between zero to 0.5) and RsTOA-method (for transmissivity greater than 0.5). However this hypothesis needs to be tested further. We shall discuss this explicitly in the revised version of the manuscript.

**Page 16**

34. L1. Briefly define what RsTOA-based method is, what is Rs method.

    **Response:** Rs-TOA-based method is the upscaling method based on $R_S$TOA and $R_S$ method is the method based on Rs. The meaning $R_S$TOA and Rs were earlier defined in the manuscript; please see Page 3 (L25 – L29). We shall further expound on it in the revised manuscript.

35. L4. $ET_{d\_pred}$ are defined early in the manuscript, consider the summary as an independent section. Better not to use these acronyms, or re-define it.

**Response:** Agreed, necessary changes will be incorporated

36. L21-25. This paragraph belongs to results & discussion section.

**Response:** Necessary changes will be incorporated

**References**

Loew, A., Peng, J., & Borsche, M. (2016). High-resolution land surface fluxes from satellite and reanalysis data (HOLAPS~v1.0): evaluation and uncertainty assessment. *Geoscientific Model Development*, *9*(7), 2499–2532. article. http://doi.org/10.5194/gmd-9-2499-2016

Malek, E. (1992). Night-time evapotranspiration vs. daytime and 24h evapotranspiration. *Journal of Hydrology*, *138*(1), 119–129. article. http://doi.org/http://dx.doi.org/10.1016/0022-1694(92)90159-S

Tolk, J, A., Howell, T. A., & Evett, S. R. (2006). Nighttime Evapotranspiration from Alfalfa and Cotton in a Semiarid Climate. *Journal of Agronomy*, *98*(3).

---

## Author Comment (AC2) · 13 Oct 2016

**Reviewer 2 (R2)**

1. How do you pick the training sites? Will the vegetation type and climate type (seasonal climate) have any effect on your trained ANN algorithm? Given that Fluxnet sites at least in N. America are mostly forest sites, will that have any potential impact on your trained ANN?'

   **Response:** The training sites were randomly selected with a representative across latitude 0-90° North and South at 10 degree interval. The vegetation type seems to have an effect on the model prediction which is already shown in Fig (9).

2. I think a paragraph on Rs and factors affecting Rs is missing from the paper. This is necessary to justify your choice of inputs for your ANN.

   **Response:** Necessary discussions will be incorporated**.**

3. Please include discussion on why the method performs poorly over cropland (Figure 9)

   Response: The probable reason of the poor $ET_d$ prediction in the croplands could be due to the effects of irrigation that is unaccounted in $ET_i$ upscaling. Since the upscaling factor is based on the ratio of instantaneous to daily shortwave radiation, the impacts due to irrigation cannot be capture, and higher errors can be expected. We shall add this description in the revised manuscript

4. As discussed in lines 25-27, $R_{sd}$ and cloudiness are directly related. ANN has no input related to cloudiness. However, you argue that you assess the performance of ANN under cloudy sky condition based on simple cloudiness index. Please elaborate on this and include discussion in the paper. Can you use Precipitation or the index of cloudiness as an input to your ANN?

   **Response:** The daily cloudiness index was estimated as the ratio between observed $R_{Sd}$ and extraterrestrial shortwave radiation to assess the performance of the ANN under variable cloud conditions. We shall add the necessary details in the discussion.

   The use of daily precipitation and soil moisture can be an improvement in the ANN model, which needs to be tested further. We shall include an analysis using a subset of sites over which daily soil moisture and rainfall data were available (as also proposed in response to R1).

5. Since vegetation plays an important role in Evapotranspiration, it would be interesting to compare different scaling methods against the type of vegetation as well (in graphs or figures)

   **Response:** We agree, and will add a comparison statistics of different scaling methods across different vegetation types.

---

## Author Comment (AC3) · 13 Oct 2016

**Reviewer 3 (R3)**

**R3 overall view on the manuscript**

 "I don't see the point of upscaling $ET_i$ to $ET_d$ for days where instantaneous observations in the optical domain are not available from satellite platforms: instantaneous $ET_i$ estimates are usually produced with instantaneous data in the optical domain, typically Thermal Infra Red data, and are therefore not computed for low transmissivities, airborne platforms excepted.

**Response: We disagree** with R3 here. R3 should be aware that there are established ET modeling schemes that explicitly considers cloudy sky cases e.g., ALEXI model,(Anderson et al., 2007). Also to overcome the cloudy sky $ET_i$ retrieval in optical domain, modeling schemes have been suggested to combine both optical and microwave remote sensing (Kustas et al., 1998). Therefore, R3's argument on ignoring $ET_i$ computation for low atmospheric transmissivities is not substantiated.

Days with low instantaneous (10AM, 1:30PM) transmissivities should be left out of the study i.e. the study should restrict to clear sky conditions from either MODIS cloud mask or, better, geostationary information (the CERES algorithm mentioned here). I therefore doubt that there is any use of the method for "Remote sensing applications" as mentioned in the title, except for UAV applications."

**Response: We do not agree** for the reasons mentioned in the previous response. The bigger picture here is focussing on the conceptual development of a robust method for upscaling $ET_i$ to $ET_d$ from remote sensing platforms across variable sky conditions that can be used for operational purpose. For remote sensing applications, the greatest challenge is the $ET_i$ upscaling in cloudy conditions, which the proposed method is able to tackle relatively better as compared to $R_STOA$ or EF based method (Table 2). R3's inclination on clear sky cases and rejecting the present method could only be applicable in predominantly pristine clear sky. We have already demonstrated this fact in Table 3 that when the temporal frequency of the data is coarse (8-day to annual), there is practically no difference between Rs and RsTOA based upscaling. But this does not deviate from the central message that Rs-based method appears to perform better when atmospheric transmissivity is between zero to 0.5.

Even for clear sky conditions the ANN method shows worse performances than the classical method based on the sole earth-sun geometrical parameters.

**Response:** It is surprising to see R3's constrained judgement on the ANN method. R3's comment on worse performance appears to be an over-statement if we consider Table 2, where MAPD between Rs and RsTOA differs by only 2-3 percent at transmissivity level above 0.5. Contrarily, we see this as an opportunity for a hybrid modeling scheme to upscale $ET_i$ across variable sky conditions by using ANN for transmissivity level of zero to 0.5 and using RsTOA method for transmissivity level above 0.5. Also, as mentioned in the manuscript, if upscaling is done from cloudy instances for a predominantly clear day, the discrepancy between ANN and RsTOA method seems to be obvious. This problem can also be overcome by including daily rainfall and soil moisture in the ANN framework. However, such hypothesis needs to be tested further. We shall add an explicit discussion on this matter in the revised version of the manuscript.

ETR between 2 successive clear sky days is an interpolation problem (which could be also treated using ANN) which needs to be tackled also.

Response: This manuscript discussed about a potential $ET_i$ upscaling strategy to convert satellite retrieved $ET_i$ to $ET_d$. We do not foresee any interpolation problem that needs to be tackled.

**R3 main comments**

1. **I also share the main concern with R1** about Energy Balance Closure: Lack of EBC should not be overlooked and is simple to correct for FLUXNET sites; it could explain the poor performance of the Evaporative Fraction method. Disregarding EBC is a major methodological flaw of the paper.

   **Response:** We propose to include an additional analysis on the performance of the three $ET_i$ upscaling methods after closing the surface energy balance in the FLUXNET sites.

2. **As criticized also by R1**, Crops and semi-arid or even dry sub humid sites are underrepresented in the FLUXNET database; this should be more carefully commented. It adds up to my concern above about the practical application of the method: TIR based daily ETR computation algorithms are particularly needed for water use monitoring in water depleted environments, much less for natural vegetation in temperate climates.'

**Response:** Under-representation of crops and semi-arid sites in the FLUXNET database does not necessarily limit the practical applications of this method. As already described in the response of R1 that the relatively high errors in $ET_d$ in croplands might be due to neglecting the irrigation effects in the ANN and inclusion of daily soil moisture and rainfall in the ANN might improve the predictive power of the modeling framework particularly over the irrigated agroecosystem. However, the performance of the method in the semi-arid shrublands appear to be promising (Fig. 9) and therefore the method seems to be credible under water-stressed environment also. This approach is equally important for natural systems e.g., in the Amazon basin or in the forest ecosystems where significant hydrological and climatological projections are emphasizing the role of $ET_d$ to understand the resilience of natural ecosystems in the spectre of hydro-climatological extremes (Harper et al., 2014; Kim et al., 2012).

3. Are the validation and the training datasets from different years? It seems to me that this is a requirement to use the method for future applications.'

   **Response:** Yes, the training and validation datasets are from different years. The validation was performed over independent sites also which are clearly delineated in Fig. 3.

4. What is the true added value of the ANN for future operational applications of the upscaling algorithm, say for an operational satellite product? This aspect, although the original motivation of the paper, is somewhat overlooked in the discussion section.'

   **Response: Yes, the true added value of the ANN is for an operational daily $ET_d$ product from polar satellites.** Currently, the polar Earth orbiting satellites provide us with $ET_i$ only. However, for most hydrological and ecosystem modeling applications, $ET_d$ is needed. Therefore, for studies that will opt to apply the Rs method as a scaling algorithm, $R_{sd}$ will be easily available for any measurement of $R_{Si}$ by the satellite using the ANN. We shall make this point explicit in the revised version of manuscript.

5. For cloudy conditions the ETR upscaling method using instantaneous solar radiation as part of the training (even from another site) performs slightly better than that based on the sole TOA solar radiation: is it mostly due to the fact that the ANN adds information on actual incoming radiation obtained at a "nearby" FLUXNET location?'

**Response: This is not true. From Table 2, it is clearly seen that the ET upscaling method based on shortwave radiation has outperformed the TOA-based method under cloudy to moderately clear sky conditions when atmospheric transmissivity is between zero to 0.5. However under the clearest sky, the shortwave radiation based method showed relatively higher RMSE than the TAO-based method. If the** ANN adds information on actual incoming radiation obtained at a "nearby" FLUXNET location, then we would expect the ANN to produce lower RMSE for all the classes of atmospheric transmissivity. These statistics rather strengthens the fact that if upscaling is done from a cloudy instance for a predominant clear sky day, higher errors can be expected from the shortwave radiation based upscaling method. We shall highlight this fact in the discussion of the revised manuscript.

**R3 Minor comments**

6.  In introduction one should add a review of which upscaling support variables can be derived from remote sensing data directly, which can be obtained indirectly from either RS data or any other distributed routinely produced data and those not obtainable from remote sensing or other distributed operational datasets.

    **Response:** Good point. We shall add few sentences on it.

**7.** How do you manage night-time conditions?'

    **Response:** The answer to this question is already provided in the response of R1.

8.  Move P5L1-4 to the end of this section and precise the variables fed by ANN upfront there.

    **Response:** Agreed.

9.  It is not clear, why there is a testing dataset and a separate validation dataset within the training dataset?'

    **Response:** The ANN algorithm is designed to validate its performance for any given training which in most cases should be sufficient for validating the network. However to ensure the network is robust, we further test the generated network with independent dataset. We shall mention this in the revised manuscript.

10. P9L5: 'Why use transmissivity rather than the ration between actual and theoretical clearsky radiations to separate the various cloudiness bins? (in order at least to separate winter conditions with lower clear sky transmissivity from summer conditions).

**Response: We disagree. Transmissivity gives the actual sky conditions and should be used to classify differential cloudiness levels. The estimation of theoretical clear-sky radiation is based on the assumption of clear sky transmissivity (which is typically 0.75). Separating sky conditions based on actual and theoretical clear sky radiation might produce baffling results in cases when actual radiation is higher than the theoretical clear sky radiation.**

11. P14L10: "would likely": this can be checked, is it the case ?'

**Response:** We shall clarify this in the revised manuscript.

12. P13L12: "reasonable" > "reasonably"

**Response:** Necessary correction will be incorporated in the revised manuscript.

**References**

Anderson, M. C., Norman, J. M., Mecikalski, J. R., Otkin, J. A., & Kustas, W. P. (2007). A climatological study of evapotranspiration and moisture stress across the continental United States based on thermal remote sensing: 1. Model formulation. *Journal of Geophysical Research: Atmospheres*, *112*(D10), n/a--n/a. article. http://doi.org/10.1029/2006JD007506

Harper, A., Baker, I. T., Denning, A. S., Randall, D. A., Dazlich, D., & Branson, M. (2014). Impact of Evapotranspiration on Dry Season Climate in the Amazon Forest. *Journal of Climate*, *27*(2), 574–591. article. http://doi.org/10.1175/JCLI-D-13-00074.1

Kim, Y., Knox, R. G., Longo, M., Medvigy, D., Hutyra, L. R., Pyle, E. H., … Moorcroft, P. R. (2012). Seasonal carbon dynamics and water fluxes in an Amazon rainforest. *Global Change Biology*, *18*(4), 1322–1334. article. http://doi.org/10.1111/j.1365-2486.2011.02629.x

Kustas, W. P., Zhan, X., & Schmugge, T. J. (1998). Combining Optical and Microwave

Remote Sensing for Mapping Energy Fluxes in a Semiarid Watershed. *Remote Sensing of Environment*, *64*(2), 116–131. article. http://doi.org/http://dx.doi.org/10.1016/S0034-4257(97)00176-4

---

## Author Response (AR1)

**Editor Decision: Publish subject to revisions (further review by Editor and Referees) (20 Oct 2016) by Miriam Coenders-Gerrits**

**Comments to the Author:**

The authors present a study where they test an ANN to upscale instantaneous remote sensing observations (Rsi, RsiTOA, RSdTOA, theta_z, and L_D) to daily Rsd estimates from where they estimate daily ET. These results are also compared to two other methods for converting instantaneous observations to daily ET estimates. The paper is well written and easy to read. Two reviewers were mainly positive, and the 3rd reviewer expressed some concerns on the validity/usefulness of the method during (partly) overcast days. I think the authors correctly replied to comments of the 3 reviewers and the proposed changes are OK. However, all 3 reviewers commented on the selection of the FLUXNET sites. How representative are the sites for different climates, biomes and time of the year/seasonality?? Although the authors replied to Reviewer #1 that they will elaborate on it, but that they already showed that it does not influence the training of the ANN, I think the study will benefit from a proof of this claim. Especially, since the main objective of the paper is to show the use of ANN for upscaling from instantaneous to daily. Therefore, I agree with the suggestion of Reviewer #1 to do a 'sensitivity' analysis for the selection of the sites in place, time and biome.

Response: A sensitivity analysis is now performed to assess the applicability of the ANN-based modeling framework to multiple biomes. The results are discussed in section 3.5.

**Minor comments:**

(1) P3L2 and L12: what is the need of using E_T and ETd? Are they not the same?

Response: By *ET*, we mean evapotranspiration, which is generic. $ET_d$ signifies daily *ET* and $ET_i$ signifies instantaneous *ET*. This uniformity is maintained throughout the manuscript.

(2) P3L3: all symbols in text in italic (throughout manuscript)

Response: Done as suggested.

(3) P4L11: "... variables (e.g., dialy..." (add comma).

Response: This sentence is modified as follows (p4, l15 to l19):

Although the $EF_r$-based method produced comparable $ET_d$ estimates as the $R_S$-based method, however the dependence of $EF_r$ estimates on certain variables (e.g., daily net available energy; $\phi$ and wind speed) and the difficulty to characterise them at the daily scale from single acquisition of polar orbiting satellites (Tang et al., 2015) makes it a relatively less attractive method.

(4) P4L11: theta is not explained.

Response: It is net available energy, explained now (p4, l17).

(5) P5L2: better: ".. predict Rsd based on Rsi satellite observations"

Response: Corrected (p5,l31).

(6) P5L2-4: Objectives 2 and 3 are not really objectives of this paper (since this is already done in the past). It's more that the results of the ANN are used to apply one method to upscale instantaneous observations to ETd and that these outcomes are then compared to two other upscaling techniques.

Response: Objective is now moved to the end of the introduction (p5 l30 − l31; p6 l1 − l3). Objectives are corrected as follows,

The objectives of the present study are: (1) using a ANN with Multilayer Perceptron (MLP) architecture to predict $R_{Sd}$ based on $R_{Si}$ satellite observations, (2) applying $R_{Sd}/R_{Si}$ ratio as a scaling factor to upscale $ET_i$ to $ET_d$ under all sky conditions, and (3) comparing the performance of proposed $R_S$-based $ET_i$ upscaling method with $R_STOA$ and $EF$-based $ET_i$ upscaling methods across a range of temporal scales, biomes and variable sky conditions.

(7) P6L7-10: What is the use of having ET with the units MJ/m2/d and Rsd in W/m2? Please use one of the two for both.

Response: Necessary corrections are done (p6, l10 − l14).

(8) P7L12: add space between (Rye et al, 2012) and mainly.

Response: Corrected (p8, l21).

(9) P7L18: What is PURELIN? Please briefly explain.

Response: Explained now (p8, l28 – l31).

(10) P8L23-25: I would link here to figure 1 and use the same letters for the 3 method. Thus a=Rs-method, b= RsdTOA-method and c=EF-method.

Response: Necessary corrections are made (section 2.4, p10, l7 – l17).

(11) Eq5: combine into 1 equation, Eq6: combine into 1 equation

Response: Now all the equations are numbered individually (section 2.4 and 2.5, p10, p11).

(12) P9L13-16: Use here the same order as the order of Eq8-12

Response: Corrected now (section 2.5, p11, l9 – l11).

(13) P10L10: Unclear/vague sentence. Please rewrite.

Response: Corrected now (p12, l17 – l18).

(14) P10L13: suggestion: use time-of-day instead of time-of-daytime.

Response: Corrected throughout in the text.

(15) P11L2: The categories of Tau are not explained in the text. When is something belonging to Tau_1 and when to Tau_4? (like explained in the caption of fig 5.)

Response: The categories of Tau is already explained in section 2.4 (p11, l2 – l5). For clarity, we again explain it (p12, l29).

(16) Table 2: Maybe make it more clear that "Rs, RsTOA and EF" are the 3 methods and not that e.g., the R2 refers to the performance of an estimation of Rs.

Response: This is now made explicit in the caption of Table 2.

(17) Table 1, 2,3: Maybe convert these tables into similar graphs like figure 11.

Response: We would prefer to keep Table 1, 2, 3 as they are in the manuscript. Representing all of them in figures similar to Fig. 11 might add monotony.

Fig 1: What is the difference between Rsd_pred and Rsd?

Response: RSd_pred is the predicted RSd from RSi. Here RSd is the generic symbol to signify daily shortwave radiation. We made the necessary correction in caption of Figure 1 caption.

Fig 5-caption: "..between Rsd_obs versus Rsd_pred...". Furthermore, also explain the transmissivity classes in the main manuscript text.

Response: Corrected accordingly.

Fig 10: this figure is hard to read. Improve quality.

Response Corrected now.

**Reviewer 1 (R1):**

1. **Energy budget closure problem at FLUXNET**. Energy budget imbalance has long been identified at FLUXNET sites. The imbalance is about -40% - +20%, indicating latent heat/sensible heat fluxes might be underestimated by up to 40%. Indeed, the energy imbalance is an existing fact we have to accept, I guess there is little can be done to overcome it in this particular study.

   **Response:** Good point. We have now included an intercomparison of $R_S$-based $ET_i$ upscaling results including both energy balance closure and non-closure in the revised version of the manuscript (p20 [section 3.4], Table 4).

2. But my concerning is: if an ANN model is trained by FLUXNET data, how much confidence do we have when we apply it to satellite retrieval? The energy budget close problem affects the results in two ways: (1) the overall robustness of the proposed upscaling method (Rs method); (2) comparison of Rs method with the evaporative fraction based upscaling (EF method Eqn. 5). However, the exo-atmospheric irradiance method is not affected (Eqn. 6). I guess the authors must be aware of this issue; it would be better to literally discuss them in the results section.

   **Response:** Regarding R1's concern on the impact of surface energy balance closure on the performance of $ET_d$ evaluation, **it is important to mention that the implicit assumption in remote sensing based $ET_i$ retrieval is the closure of surface energy balance**. Therefore, for the remote sensing retrievals, the energy balance closure problems will not affect $ET_d$ estimates in the current framework of ANN. **However, for the validation of remote sensing based $ET_d$ retrievals, surface energy balance fluxes from eddy covariance measurements need to be closed. This is now mentioned in section 3.4** (p20 [section 3.4])

   In the present study, the closure problem of surface energy balance will affect the evaluation statistics of all the three methods, and therefore, we included an intercomparison of $R_S$-based $ET_i$ upscaling results including both energy balance closure and non-closure in the revised version (Table 4, section 3.4). **As compared to the *EF* and $R_S TOA$ approach, the $R_S$-based method is more robust with regards to *ET* scaling on**

**a daily time frame since the method carries maximum information on the cloudiness, which is a key limiting factor in upscaling of $ET_i$ to $ET_d$.**

With reference to Eq. (1), the network developed is intended to develop an operational method to directly upscale $ET_i$ (estimated from polar orbiting satellites) to $ET_d$ based on the ratio of daily to instantaneous shortwave radiation ($R_{Sd}$ and $R_{Si}$). Given there is no direct method to directly estimate $R_{Sd}$ from remote sensing satellite, we trained an ANN with the FLUXNET observations of $R_{Si}$ and $R_{Sd}$, and validated the model to predict $R_{Sd}$ over independent sites, followed by using $R_{Sd}/R_{Si}$ ratio to convert $ET_i$ to $ET_d$. The datasets used for the ANN development covers a wide range of biome, climate, and variable sky conditions. Therefore, we assume the $R_{Sd}$ prediction from ANN to capture a broad spectrum of radiative forcing, which is also reflected in the independent validation of $R_{Sd}$ and $ET_d$ (Fig. 5, Fig. 7, Table 2). The performance of this model for satellite retrieval of $R_{Sd}$ (from $R_{Si}$) is dependent on the accuracy of $R_{Si}$ retrieval (Loew, Peng, & Borsche, 2016). We have discussed these in the conclusion section (p21, l10 – l26). Also, the distribution of sites over the tropics, Africa, and SE Asia are poor, and more sites over these regions are expected to make the ANN model more robust, which is mentioned in the revised manuscript (p21, l24 – l26).

Regarding R1's concern on the robustness of the approach, we have performed a sensitivity analysis of $R_S$-based ANN performance by training ANN with data from different biome combinations and compared $ET_d$ prediction statistics of the different combinations (section 3.5, Figure 13).

3. **Cloudy-sky issue**. The biggest problem of the proposed upscaling method (Rs method) is that the ANN model does not include any information about "cloudiness". Therefore, model performance under cloudy-sky condition (or low atmospheric transmissivity) is much worse than clear-sky condition. One way to tackle it, is to use climatology precipitation data. Rainfall (highly related to cloudiness) has seasonal pattern, at least for some regions (e.g., tropical rainforest, savanna). Similarly, dry season-wet seasons could provide ANN model with additional information about "possibility" of the "cloudy-sky condition" during a certain time period. In Figure 7, the overestimation of ET under cloudy sky condition is "systematic", meaning there might be a simple way to

"systematically" down regulate the ET as long as the ANN model knows it's a cloudy day.

**Response**: Good suggestion indeed. Following R1's suggestion, we tested this hypothesis by including the precipitation and soil moisture information with $R_S$ and trained a new ANN to evaluate if the inclusion of precipitation and soil moisture improved the performance of $ET_d$ prediction under persistent cloudy-sky conditions. This shows substantial improvement in $ET_d$ prediction under cloudy-sky cases [section 3.2 (p15, l1 – l27) (Figure 9)].

Including cloudiness as an input variable of the network during training process would significantly enhance the performance of the network. Use of daily precipitation and soil moisture as an indicator of cloudiness would have been the most appropriate approach in this circumstance. However the cloud information available from alternative sources e.g. from the Clouds and Earth's Radiant Energy System (CERES), the International Satellite Cloud Climatology Project–Flux Data (ISCCP-FD), and Global Energy and Water cycle Experiment Surface Radiation Budget (GEWEX-SRB) are available at coarse spatial resolution and there will be a scale mismatch. However, the precipitation data was not consistently available for most of the sites and the data gaps were significant to alter the sampling sizes. However for future studies, including cloudiness or daily precipitation as a variable in the training of the ANN to predict $R_{Sd}$ is highly recommended. On the issue of systematic errors as a result of cloud conditions, we certainly expect overestimation or underestimation. The results are discussed in section 3.2 (p15, l16 – l27) (Figure 9).

4. **FLUXNET site selection**. It was stated that the partition of data into training and validation was randomly selected. However, it's not clear whether the selected training sites are represent it cover a full range of (from dry to wet) rainfall regimes? For each vegetation type, how much percentage of data is selected to train the model? FLUXNET has more forest sites than grass/shrub sites. Are grass/shrub sites less represented in the training dataset? Following question: is the ANN model sensitive the FLUXNET site selection? This could be evaluated by doing e.g., 10 ensemble of random selection of FLUXNET sites. And check the difference among the resultant 10 ANN models?

**Response**: Since this analysis was based on FLUXNET sites distributed across 0-90
degrees latitude north and south, the training datasets covers substantial climatic and
vegetation variability. The percentage distribution of the training data according to
vegetation type was; 23% crops, 31% deciduous broadleaf forest, 10% evergreen
broadleaf forest, 20% evergreen need leaf forest, 8% grassland, 7% shrubs and 1% aquatic
as indicated in table S1. The number of grassland and shrubs as indicated were relatively
less as compared to the crops and forests sites. **However, biome specific error statistics**
**(Fig. 11) indicated the absence of any systematic errors due to vegetation sampling**
**with the exception of EBF. Availability of more EBF sites in the training datasets is**
**expected to reduce the cloudy sky errors substantially. We have elaborated this**
**discussion in the revised manuscript.**

We have also performed a sensitivity analysis of $R_S$-based ANN performance by randomly
training ANN with data from different biome combinations and compared $ET_d$ prediction
statistics of the different combinations (section 2.6). The results are discussed in section
3.5 (p20, l22 – l30 and p21, l1 – l8) (Figure 13).

5.  **Crop ET.** I think the proposed method might be only suitable for estimating natural
terrestrial ecosystem *ET*. There is large bias of crop *ET* estimation (Figure 9). That could
be due to irrigation? Land management? Those anthropogenic factors (largely alter land
surface water budget) is not included in the ANN model and the *ET* estimation.

**Response:** Figure 9 is now Figure 11.

Both the current framework and *RsTOA*-based method of $ET_d$ estimation would be best
suited for natural ecosystem as well for the rainfed agroecosystems. In the biome specific
$ET_d$ error statistics (Fig. 11), relatively large bias in crop *ET* is propagated due to the
inclusion of irrigated agroecosystems in the validation. Inclusion of daily soil moisture
and rainfall in the ANN has shown to improve the $R_S$-based $ET_d$ prediction only under
persistent cloudy-sky conditions. In irrigated agroecosystems, day-to-day variation in soil
moisture is not substantial and evapotranspiration is predominantly controlled by the net
radiation. Therefore, the inclusion of soil moisture and rainfall in the current ANN
framework had not made any improvement in the $ET_d$ prediction statistics in irrigated
agroecosystems. Further, having many explanatory variables (e.g., land management, irrigation statistics, anthropogenic factors) to train the ANN, we risk overfitting the model and hence introducing bias. There are now described in the revised manuscript (section

3.3, p17, l10 – l24).

6. **Vegetation control on ET.** The proposed upscaling method is based on the idea that higher available energy (Rs) lead to higher evapotranspiration (*ET*) (Eqn. 1). It basically assumes that the Bowen ratio does not change during the daytime, so that instantaneous

ET/Rs is equal to daily ET/Rs. However, it ignores the important fact that *ET* is also mediated by vegetation via stomata control. For example, trees and grass have dramatically different stomata density, stomata size. Therefore, their stomata open/closure and its control on water vapor conductance are different.

**Response:** This is indeed a very good point and is discussed in section 4 (p22, l10 – l24)

of the manuscript. The proposed upscaling method is based on the idea that instantaneous

$ET/R_S$ is equal to daily $ET/R_S$, although it implicitly includes the stomatal controls on $ET$

observations mediated by the vegetation. The cases where $ET_i$ is low due to water stress induced strong stomatal control; low magnitude of $ET$ will also be reflected in upscaling

$ET_i$ to $ET_d$ (according to eq. 1). However, to account any carry over effects of the stomatal control on $ET_d$, inclusion of longwave radiation would likely to improve the scheme.

Stomatal control is significantly dependent on the thermal longwave radiative components, and, therefore, the relative proportion of downwelling and upwelling longwave radiation is expected to be a stomatal constraint. However, the availability of longwave radiation measurement stations in the FLUXNET datasets is limited to formulate ANN and evaluate this hypothesis. In general, the stomatal and biophysical constraints are imposed in state-of-the-art thermal remote sensing based $ET_i$ retrieval schemes, and, therefore the ANN framework can be applied to upscale remote sensing based $ET_i$ to $ET_d$. Also, relatively good performance of the model in semiarid shrubland also indicated the applicability of the method in water stressed ecosystems where stomatal controls are predominant.

7. The question is: it is worthwhile to add biome type information in the ANN model? Is it possible to further improve the results (Figure 9) for forest sites by considering biome type information in the ANN model and *ET* estimates?

**Response:** It is not worthwhile to add biome type information in the ANN model. The performance of ANN is principally dependent on atmospheric radiative forcings and less on biome types. To test this hypothesis, we have also performed a sensitivity analysis of $R_S$-based ANN performance by randomly training ANN with data from different biome combinations and compared $ET_d$ prediction statistics of the different combinations (section 2.6). The results are discussed in section 3.5 (p20, l22 – l30 and p21, l1 – l8) (Figure 13).

**Minor comments**

**Page2**

8. L4. a key challenge in mapping regional *ET* using polar orbiting sensors

**Response:** Necessary changes are incorporated (p2, l3 – l4).

9. L6. On the terrestrial surface -> remove

**Response:** Removed.

10. L8. The approach relies on : : : -> remove

**Response:** Removed.

11. L16. derived from simple mathematical computation -> replace: e.g., solar zenith angle, day length

**Response:** Changes are made as suggested (p2, l13 – l14).

12. L20. Based on the measurements from 126 sites -> remove

**Response:** Removed.

13. L20. Rs-based upscaling produced

**Response:** Necessary changes are incorporated (p2, l17)

14. L7. ET variability is influenced by (1) available energy received, (2) soil moisture supply and (3) vegetation mediation. I think the third one is missing here. To be complete, the three key factors should all be fairly discussed in the introduction

   **Response:** Good point. We included the vegetation controls on ET in the introduction (p3, l11 – l12).

15. L9. "Therefore" is not appropriate here, there is no cause-effect relationship here. Better start a new paragraph and discuss the major challenges in Et upscaling

   **Response:** Done (p3, l13).

**Page4**

16. L19. Estimate $R_{sd}$ form any specific time-of-day $R_{Si}$ information. But isn't the value of this study is to predict $R_{sd}$ based at satellite local crossing time (e.g., 10:30, 13:30)?

   **Response:** The aim of this study is to help develop an approach that would help in the upscaling of $ET_i$ (retrieved at satellite overpass time) to $ET_d$. The value of this study consists of exploiting $R_{Si}$ information at satellite local crossing time to predict $R_{Sd}$ which is not directly retrievable from any polar orbiting satellites, so that the ratio of $R_{Sd}/R_{Si}$ can be further used to upscale $ET_i$ to obtain daily $ET$ ($ET_d$) estimates (in the framework of eqn. 1). Currently we are limited to demonstrating with MODIS overpass times (Terra and Aqua), however in case there are new missions in the future with different local overpass time, the method would still be applicable. This description is made explicit in the revised manuscript (section 2.1, p7, l16 – l22).

17. L22. L22. In order -> remove

   **Response:** Removed (p4, l28).

18. L24. ANN is a non-linear model. Multi-layer perceptron (MLP) is.. These sentences belong to method section.

   **Response:** The description is now moved in the beginning of section 2.2 (p8, l2 – l10).

**Page5**

19. L13. Cloudiness is a phenomenon. These sentences belong to discussion section.

**Response:** This sentence is moved to section 3.5 (p21, l3 – l4).

**Page6**

20. L6. Two question: (1) Does Eqn. 1 assume the Bowen ratio is constant during daytime? (2) Does it ignore the night time *ET*, which could be large when surface wind speed is high?

**Response:** According to eqn. 1,

$$ET_d/ET_i \approx R_{Sd}/R_{Si}$$

**and**

$$ET_d/ET_i = EF_d(R_N - G)_d/EF_i(R_N-G)_i$$

Where EF is the evaporative fraction, $R_N$ is net radiation, and G is ground heat flux.

Therefore, eqn. 1 is based on the assumption that shortwave radiation is the principal driver of evaporative flux. Although *ET* can be limited due to both radiation and water, but in the water limited ecosystems the magnitude of $ET_i$ will also be low due to low soil moisture availability and therefore and upscaling $ET_i$ to $ET_d$ in the framework of eqn. 1 may not introduce significant error. The evidence is already seen in Fig. 9 where shrublands showed relatively lower RMSE (despite being water limited) as compared to the forests. We have added this discussion in the revised manuscript (section 2.1, p6, l14 – l24).

 (2) The analysis is based on 24-hour period, meaning night time *ET* contribution is implicitly considered. However, studies have ready shown that the nighttime ET in semi-arid regions contributes only $2 – 5\%$ of the total season *ET* (Malek, 1992; Tolk, J, Howell, & Evett, 2006), and therefore does not appear to be significant. This is mentioned in section 2.1 (p7, l12 – l15).

**Page8**

21. L16. In a percentage ratio of 80:15:15. Is this right? Shouldn't be 80:15:5 or 70:15:15?

**Response:** The ratio should be 80:15:5, corrections are made in the revised manuscript (p9, l20).

22. L9. We first evaluate the efficacy of the ANN method for predicting $R_{sd}$.

**Response:** Necessary changes are incorporated (p12, l18).

23. L12. As obtained following the methodology described in the section 2.1 -> remove

**Response:** Necessary changes are incorporated (p12, l19)

24. L13. Showing -> including

**Response:** Necessary changes are incorporated (p12, l19)

25. L14. From the analysis it is apparent that -> remove

**Response:** Removed (p12, l20).

26. L1. Figure 5 evaluates the $R_{sd\_pred}$ under different level of clear sky transmissivity

**Response:** Necessary changes are incorporated in the revised manuscript (p12, l28).

27. L3. What if the ANN model includes "clear sky transmissivity, would model performance under cloudy sky condition be improved?

**Response:** We do not think so, because including clear sky transmissivity could make the modeling framework biased towards clear sky cases only.

28. L16. Using $R_{sd\_pred}/R_{si}$ as a scaling factor following eq. 1 -> remove

**Response:** Necessary changes are incorporated (p13).

29. L1. Figure 7 compares $ET_{d\_pred}$ against $ET_{d\_obs}$ for different level of daily. The overall RMSE, MAPE

**Response:** Necessary changes are incorporated ($p14, l1$).

30. L4. Given that the overestimation is a systematic, is it possible to eliminate it or reduce it? The overestimation was due to the fact that during the specific time slot of interest (e.g., 11:30) the sky is clear while the sky is cloudy during other times. However, there could be another opposite case that sky is cloudy at e.g., 11:30 but clear at other times. It will probably lead to an underestimation of $R_{Sd\_pred}$, and consequently underestimation of $ET_{d\_pred}$. I am wondering why the latter is not the case at least in Figure 7.

**Response:** This is a very good argument. With the current framework of ANN, this systematic overestimation cannot be eliminated. However, as demonstrated in Fig 11, with the inclusion of daily rainfall and soil moisture in the ANN model, such overestimation tendency could be reduced ($p15, l1 – l9$).

Regarding R1's argument on finding underestimation of $ET_d$ from 1100 hr cloudy sky $ET_i$ upscaling in a predominant clear day, such cases were also found in $\tau_3$ category (Fig. 7) where clouds of data points clearly falling significantly below the 1:1 line, thus showing substantial underestimation of $ET_d$. We have included this discussion in the revised manuscript ($p14, l11 – l15$).

31. L14.higher errors in $ET_{d\_pred}$ can be expected. Is there a way to overcome this problem?

**Response:** One of the probable ways to overcome the errors in cloudy sky is to incorporate daily rainfall and soil moisture or information of cloud cover in the ANN. This is now demonstrated in the revised manuscript and related discussions are included in ($p15, l1 – l11$).

32. L24. Again, biome specific results are related to the clear-sky issue. Tropical evergreen broadleaf forests have high ET, water tends to re-cycle locally and generate rainfall. It's reasonable to see that cloudy sky condition is more frequent at tropical evergreen broadleaf forest than e.g., at grass land.

Response: Agreed. This point is added in the discussion of the revised manuscript. This discussion is now moved in section 3.3 ($p16, l29 – l30; p17, l1 – l24$).

33. L27. ET estimations at cropland were much worse than grass. It that because e.g., irrigation? Land management? Or any other anthropogenic factors that are not considered in the ANN model? Page 13.

Response: Yes, the farm management practice especially irrigation might have impact on the output for example in a case where irrigation was carried out for three consecutive days yet the sky conditions were consistently cloudy would present a challenge. Necessary discussions are in section 3.3 (p17, l16 − l18).

**Page 13**

34. L20. Based on Table 2, Figure 11, $R_S$TOA method seems successful. Under clear sky condition, it was even better than the proposed Rs method. Further, over longer time scale (annually), there is no big difference between $R_S$TOA and Rs.

Response: Agreed and discussed also in the manuscript. As shown in Table 2, relatively lower RMSE of RsTOA for atmospheric transmissivity class above 0.75 reveals that under pristine clear sky conditions RsTOA can be successfully used to upscale $ET_i$. However, one of the main reasons for the differences in RMSE between Rs and RsTOA method for daily transmissivity above 0.75 could be due to the fact that if $ET_i$ upscaling is performed from a cloudy instance for a predominantly clear sky day, then such RMSE difference between the two different upscaling methods is expected. These results also showed the probability of a hybrid $ET_i$ upscaling method by combining Rs-method (for transmissivity between zero to 0.5) and RsTOA-method (for transmissivity greater than 0.5). However this hypothesis needs to be tested further. Discussions are included in section 3.3 (p18, l23 − l30, p19, l1 − l2) of the revised manuscript.

**Page 16**

35. L1. Briefly define what RsTOA-based method is, what is Rs method.

**Response:** Rs-TOA-based method is the upscaling method based on $R_S$TOA and $R_S$ method is the method based on Rs. The meaning $R_S$TOA and Rs were earlier defined in the manuscript; please see Page 4 (l1 − l5). We have further expanded this in the conclusion section (p21).

36. L4. $ET_{d\_pred}$ are defined early in the manuscript, consider the summary as an independent section. Better not to use these acronyms, or re-define it.

**Response:** Agreed, necessary changes are made (p22, l4 – l5)

37. L21-25. This paragraph belongs to results & discussion section.

**Response:** This paragraph is now moved to section 3.3 (p19, l28 onwards)

**Reviewer 2 (R2)**

21. How do you pick the training sites? Will the vegetation type and climate type (seasonal climate) have any effect on your trained ANN algorithm? Given that Fluxnet sites at least in N. America are mostly forest sites, will that have any potential impact on your trained ANN?'

**Response:** The training sites were randomly selected with a representative across latitude 0-90° North and South at 10 degree interval. The potential impact of vegetation on ANN training is now described in section 3.5 (p21 – p22) through a sensitivity analysis of ANN performance to different training scenarios.

92. I think a paragraph on Rs and factors affecting Rs is missing from the paper. This is necessary to justify your choice of inputs for your ANN.

**Response:** Necessary discussions are incorporated in section 2.1 (p7, l23 – l31).

123. Please include discussion on why the method performs poorly over cropland (Figure 9)

Response: The probable reason of the poor $ET_d$ prediction in the croplands could be due to the effects of irrigation that is unaccounted in $ET_i$ upscaling. Since the upscaling factor is based on the ratio of instantaneous to daily shortwave radiation, the impacts due to irrigation cannot be capture, and higher errors can be expected. We have added this description in the revised manuscript (section 3.3, p17, l11 onwards).

184. As discussed in lines 25-27, $R_{sd}$ and cloudiness are directly related. ANN has no input related to cloudiness. However, you argue that you assess the performance of ANN under cloudy sky condition based on simple cloudiness index. Please elaborate on this and include discussion in the paper. Can you use Precipitation or the index of cloudiness as an input to your ANN?

**Response:** The daily cloudiness index was estimated as the ratio between observed $R_{Sd}$ and extraterrestrial shortwave radiation to assess the performance of the ANN under variable cloud conditions (p11, l1 – l7).

The use of daily precipitation and soil moisture can be an improvement in the ANN model. To test this hypothesis, we have included an analysis using a subset of sites over which daily soil moisture and rainfall data were available. The results are shown in Figure 9. Necessary descriptions are added in section 3.2 (p15, l1 – l27).

15. Since vegetation plays an important role in Evapotranspiration, it would be interesting to compare different scaling methods against the type of vegetation as well (in graphs or figures)

**Response:** We have added a comparison statistics of two different scaling methods (Rs-based and RsTOA-based) across different vegetation types (Fig. 11) and the results are explained in section 3.3 (p16, l29 – l30; p17, l1 – l10).

**Reviewer 3 (R3)**

**R3 overall view on the manuscript**

(1) "I don't see the point of upscaling $ET_i$ to $ET_d$ for days where instantaneous observations in the optical domain are not available from satellite platforms: instantaneous $ET_i$ estimates are usually produced with instantaneous data in the optical domain, typically Thermal Infrared data, and are therefore not computed for low transmissivities, airborne platforms excepted.

**Response: We disagree** with R3 here. R3 should be aware that there are established ET modeling schemes that explicitly considers cloudy sky cases e.g., ALEXI model (Anderson et al., 2007). Also to overcome the cloudy sky $ET_i$ retrieval in optical domain, modeling schemes have been suggested to combine both optical and microwave remote sensing (Kustas et al., 1998). Therefore, R3's argument on ignoring $ET_i$ computation for low atmospheric transmissivities is not substantiated.

(2) Days with low instantaneous (10AM, 1:30PM) transmissivities should be left out of the study i.e. the study should restrict to clear sky conditions from either MODIS cloud mask or, better, geostationary information (the CERES algorithm mentioned here). I therefore doubt that there is any use of the method for "Remote sensing applications" as mentioned in the title, except for UAV applications."

**Response: We do not agree** for the reasons mentioned in the previous response. The bigger picture here is focussing on the conceptual development of a robust method for upscaling $ET_i$ to $ET_d$ from remote sensing platforms across variable sky conditions that can be used for operational purpose. For remote sensing applications, the greatest challenge is the $ET_i$ upscaling in cloudy conditions, which the proposed method is able to tackle relatively better as compared to $R_STOA$ or EF based method (Table 2). R3's inclination on clear sky cases and rejecting the present method could only be applicable in predominantly pristine clear sky. We have already demonstrated this fact in Table 3 that when the temporal frequency of the data is coarse (8-day to annual), there is practically no difference between Rs and RsTOA based upscaling. But this does not deviate from the central message that Rs-based method appears to perform better when atmospheric transmissivity is between zero to 0.5.

(3) Even for clear sky conditions the ANN method shows worse performances than the classical method based on the sole earth-sun geometrical parameters.

**Response:** It is surprising to see R3's constrained judgement on the ANN method. R3's comment on worse performance appears to be an over-statement if we consider Table 2, where MAPD between Rs and RsTOA differs by only 2-3 percent at transmissivity level above 0.5. Contrarily, we see this as an opportunity for a hybrid modeling scheme to upscale

$ET_i$ across variable sky conditions by using ANN for transmissivity level of zero to 0.5 and using RsTOA method for transmissivity level above 0.5. Also, as mentioned in the manuscript, if upscaling is done from cloudy instances for a predominantly clear day, the discrepancy between ANN and RsTOA method seems to be obvious. This problem can also be overcome by including daily rainfall and soil moisture in the ANN framework, which is now demonstrated in the revised manuscript (section 3.2, p15, l1 – l27).

(4) ETR between 2 successive clear sky days is an interpolation problem (which could be also treated using ANN) which needs to be tackled also.

Response: This manuscript discussed about a potential $ET_i$ upscaling strategy to convert satellite retrieved $ET_i$ to $ET_d$. We do not foresee any interpolation problem that needs to be tackled.

**R3 main comments**

1. **I also share the main concern with R1** about Energy Balance Closure: Lack of EBC

should not be overlooked and is simple to correct for FLUXNET sites; it could explain the poor performance of the Evaporative Fraction method. Disregarding EBC is a major methodological flaw of the paper.

**Response:** We have included an additional analysis on the performance of the proposed

$ET_i$ upscaling method after closing the surface energy balance in the FLUXNET sites in section 3.4 (p20, l4 – l21). All the existing literatures have already demonstrated the poor performance of evaporative fraction based ET upscaling methods despite EBC closure.

2. **As criticized also by R1**, Crops and semi-arid or even dry sub humid sites are underrepresented in the FLUXNET database; this should be more carefully commented. It adds up to my concern above about the practical application of the method: TIR based daily ETR computation algorithms are particularly needed for water use monitoring in water depleted environments, much less for natural vegetation in temperate climates.'

**Response:** Under-representation of crops and semi-arid sites in the FLUXNET database does not necessarily limit the practical applications of this method. As already described in the response of R1 that the relatively high errors in $ET_d$ in croplands might be due to neglecting the irrigation effects in the ANN and inclusion of daily soil moisture and rainfall in the ANN might improve the predictive power of the modeling framework particularly over the irrigated agroecosystem. However, the performance of the method in the semi-arid shrublands appear to be promising (Fig. 11) and therefore the method seems to be credible under water-stressed environment also. This approach is equally important for natural systems e.g., in the Amazon basin or in the forest ecosystems where significant hydrological and climatological projections are emphasizing the role of $ET_d$ to understand the resilience of natural ecosystems in the spectre of hydro-climatological extremes (Harper et al., 2014; Kim et al., 2012). These are discussed in section 3.3 (P17, l7 – l31; p18, l1 – l3).

3. Are the validation and the training datasets from different years? It seems to me that this is a requirement to use the method for future applications.'

**Response:** Yes, the training and validation datasets are from different years. The validation was performed over independent sites also which are clearly delineated in Fig. 3.

**4.** What is the true added value of the ANN for future operational applications of the upscaling algorithm, say for an operational satellite product? This aspect, although the original motivation of the paper, is somewhat overlooked in the discussion section.'

**Response: Yes, the true added value of the ANN is for an operational daily $ET_d$ product from polar satellites.** Currently, the polar Earth orbiting satellites provide us with $ET_i$ only. However, for most hydrological and ecosystem modeling applications, $ET_d$ is needed. Therefore, for studies that will opt to apply the Rs method as a scaling algorithm, $R_{sd}$ will be easily available for any measurement of $R_{Si}$ by the satellite using the ANN. We have made this point explicit in the conclusion (section 4) of the revised manuscript (p22, l28 – l31; p23, l1).

**5.** For cloudy conditions the ETR upscaling method using instantaneous solar radiation as part of the training (even from another site) performs slightly better than that based on the sole TOA solar radiation: is it mostly due to the fact that the ANN adds information on actual incoming radiation obtained at a "nearby" FLUXNET location?'

**Response: This is not true. From Table 2, it is clearly seen that the ET upscaling**

**method based on shortwave radiation has outperformed the TOA-based method**

**under cloudy to moderately clear sky conditions when atmospheric transmissivity is**

**between zero to 0.5. However under the clearest sky, the shortwave radiation based**

**method showed relatively higher RMSE than the RsTOA-based method. If the** ANN

adds information on actual incoming radiation obtained at a "nearby" FLUXNET location, then we would expect the ANN to produce lower RMSE for all the classes of atmospheric transmissivity. These statistics rather strengthens the fact that if upscaling is done from a cloudy instance for a predominant clear sky day, higher errors can be expected from the shortwave radiation based upscaling method. Discussions are already included in the revised manuscript (p18, l12 – l18; p19, l1 – l2).

**R3 Minor comments**

6. In introduction one should add a review of which upscaling support variables can be derived from remote sensing data directly, which can be obtained indirectly from either RS

data or any other distributed routinely produced data and those not obtainable from remote sensing or other distributed operational datasets.

**Response:** Good point. We have added necessary description in the introduction (p4, l28 –

l31; p5, l1 – l3) and also in section 2.1 (p7, l23 – l31) of the revised manuscript.

**7.** How do you manage night-time conditions?'

**Response:** The answer to this question is already provided in the response of R1 (p7, l12 –

l15).

8. Move P5L1-4 to the end of this section and precise the variables fed by ANN upfront there.

**Response:** Agreed. The objectives are moved at the end of the introduction.

9. It is not clear, why there is a testing dataset and a separate validation dataset within the training dataset?'

**Response:** The ANN algorithm is designed to validate its performance for any given training which in most cases should be sufficient for validating the network. However to ensure the network is robust, we further test the generated network with independent dataset. We have mentioned this this in the revised manuscript (p9, l20 – l23).

10. P9L5: 'Why use transmissivity rather than the ration between actual and theoretical clearsky radiations to separate the various cloudiness bins? (in order at least to separate winter conditions with lower clear sky transmissivity from summer conditions).

**Response:** We disagree. Transmissivity gives the actual sky conditions and should be used to classify differential cloudiness levels. The estimation of theoretical clear-sky radiation is based on the assumption of clear sky transmissivity (which is typically 0.75). Separating sky conditions based on actual and theoretical clear sky radiation might produce baffling results in cases when actual radiation is higher than the theoretical clear sky radiation.

11. P14L10: "would likely": this can be checked, is it the case?'

**Response:** Corrected (p18, l11).

12. P13L12: "reasonable" > "reasonably"

**Response:** Corrected (p16, l7).

[revised manuscript text omitted]